## REPORT

# Inwardly rectifying potassium channels promote directional sensing during neutrophil chemotaxis

Tianqi Wang[1]*, Daniel H. Kim[1]*, Chang Ding[1]*, Dingxun Wang[2], Weiwei Zhang[3], Martin Silic[2], Xi Cheng[1], Kunming Shao[1], TingHsuan Ku[1], Conwy Zheng[1], Junkai Xie[4], Shulan Xiao[5,6], Krishna Jayant[5,6], Chongli Yuan[4], Alexander A. Chubykin[1,5,6], Christopher J. Staiger[1,3], GuangJun Zhang[2], and Qing Deng[1]

**Potassium channels control membrane potential and various physiological processes, including cell migration. However, the specific role of inwardly rectifying potassium channels in immune cell chemotaxis remains unknown. Here, we demonstrate that inwardly rectifying potassium channels, particularly Kir7.1 (*Kcnj13*), maintain the resting membrane potential and are crucial for directional sensing during neutrophil chemotaxis. Blocking or knocking out Kir in neutrophils disrupted their ability to sense direction toward different chemoattractants in multiple models. Using genetically encoded voltage indicators, we observed oscillating hyperpolarization during tail retraction in zebrafish neutrophils, with Kir7.1 required for depolarization toward the chemokine source. Focal depolarization via optogenetics biased pseudopod selection and triggered new protrusions, which depended on Gα signaling. Global hyperpolarization caused neutrophils to stall migration. Additionally, Kir influences GPCR signaling activation in dHL-60 cells. This research introduces membrane potential as a key component of the complex feedforward mechanism that links the adaptive and excitable networks necessary to guide immune cells in challenging tissue environments.**

## Introduction

Neutrophils play a vital role in innate immunity as the first line of defense against pathogens and as key regulators of the cancer microenvironment (Burn et al., 2021; Zhou et al., 2023). Neutrophil migration is a coordinated cycle involving protrusion formation and tail retraction, primarily controlled by the actin cytoskeleton. Signaling from G-protein–coupled receptors (GPCRs) converts shallow gradients of extracellular signals into a steep polarization of intracellular signaling molecules such as phosphoinositide 3-kinase (PI3K), the PI-3-phosphatase PTEN, and monomeric G proteins (Ras, Rho, Rac, and Cdc42), enabling the cell to interpret guidance cues (Hoeller et al., 2016) and directing neutrophil chemotaxis.

Membrane potential (MP) across the plasma membrane is a fundamental characteristic of all animal cells, which have a lipid bilayer that is impermeable to ions (Kulbacka et al., 2017). The transmembrane voltage generates a strong electric field across this thin lipid bilayer, influencing the activity of membrane proteins and solute exchange. This process is vital for numerous physiological functions, including the regulation of circadian rhythms in neurons and fibroblasts, biological sensing by neurons, and volume regulation (Abdul Kadir et al., 2018). The cell

MP, a key element of bioelectricity, controls the actin cytoskeleton and adhesion signals (Schwab et al., 2012; Schwab et al., 2008). Conversely, the actin cytoskeleton and related proteins regulate ion channel activity, creating complex feedback loops (Mazzochi et al., 2006).

Potassium channels are key players in establishing cell MP and cell migration (Schwab et al., 2012; Schwab et al., 2008). Potassium and other ion channels are vital for neutrophil recruitment and function (Immler et al., 2018). Previous ex vivo measurements using MP dyes show that the resting potential of neutrophils is around –74 mV. The chemokines fMLP and C5a cause various levels of depolarization (Fletcher and Seligmann, 1986; Messerer et al., 2018; Seeds et al., 1985). The main current in unstimulated mouse neutrophils is a constitutively active, external K⁺-dependent, strong inwardly rectifying current, likely through Kir2.1 (Masia et al., 2015). The calcium-activated potassium channel Kca3.1 is necessary for the controlled decrease in neutrophil volume and chemotaxis, without affecting calcium entry or reactive oxygen species production (Henríquez et al., 2016). The voltage-gated potassium channel Kv1.3 supports sustained Ca²⁺ influx and tight adhesion to blood vessel walls

[1]Department of Biological Sciences, Purdue University, West Lafayette, West Lafayette, IN, USA;  [2]Department of Comparative Pathobiology, Purdue University, West Lafayette, West Lafayette, IN, USA;  [3]Department of Botany and Plant Pathology, Purdue University, West Lafayette, West Lafayette, IN, USA;  [4]Davidson School of Chemical Engineering, Purdue University, West Lafayette, West Lafayette, IN, USA;  [5]Weldon School of Biomedical Engineering, Purdue University, West Lafayette, West Lafayette, IN, USA;  [6]Purdue Institute for Integrative Neuroscience, Purdue Autism Research Center, West Lafayette, IN, USA.

*T. Wang, D.H. Kim, and C. Ding contributed equally to this paper.  Correspondence to Qing Deng: deng67@purdue.edu;  GuangJun Zhang: gjzhang@purdue.edu.

during acute inflammation (Immler et al., 2022). ATP-sensitive K⁺ channels also play a role in regulating neutrophil migration (Da Silva-Santos et al., 2002). Besides K⁺ channels, anion transporters are also essential for neutrophil transendothelial migration and chemotaxis (Moreland et al., 2006; Volk et al., 2008). However, the spatial-temporal dynamics of the MP during neutrophil migration remain uncharacterized.

The inwardly rectifying potassium channel subfamily J member 13 (*Kcnj13*, which encodes Kir7.1) helps maintain the resting potential and controls the organization of smooth muscle cytoskeleton during mouse tracheal tubulogenesis (Yin et al., 2018). In humans, mutations in the *KCNJ13* gene are associated with retinal degeneration, including autosomal-dominant snowflake vitreoretinal degeneration (Hejtmancik et al., 2008), and recessive mutations lead to Leber congenital amaurosis, an early-onset form of blindness (Sergouniotis et al., 2011). In zebrafish, different mutations in *kcnj13* are associated with changes in pigment patterns and fin development (Podobnik et al., 2020; Silic et al., 2020), suggesting that this channel may play a regulatory role in cell migration. The zebrafish serves as an excellent model for studying neutrophil biology because of the high conservation of genetic, biochemical, and morphological features compared with mammals (Deng and Huttenlocher, 2012). Here, we used the zebrafish neutrophil model to explore the role of Kir7.1 in neutrophil chemotaxis.

## Results and discussion
### Kir regulates neutrophil chemotaxis
We analyzed our previously published mRNA-sequencing data on sorted zebrafish neutrophils (Hsu et al., 2019) and performed RT-PCR to confirm the expression of *kcnj1b*, *kcnj11*, and *kcnj13* (Fig. S1 A). We used VU590, a potent and moderately selective inhibitor for Kir1.1 (encoded by kcnj1) (IC50 = 290 nM) and Kir7.1 (IC50 = 8 μM) (Weaver and Denton, 2021), to investigate whether Kir regulates neutrophil migration in 3-day post-fertilization (dpf) larvae from *Tg(lyzC:mCherry)*[pu43]. Treatment with VU590 did not change neutrophil numbers (Fig. 1 A) or their spontaneous migration in fish (Fig. 1, B and C), but it significantly decreased recruitment to tail transection sites (Fig. 1, D–F). Additionally, VU590 treatment substantially reduced neutrophil chemotaxis from the caudal hematopoietic tissue (CHT) to the ventral fin fold toward the chemoattractant leukotriene B4 (LTB4) (Yoo et al., 2011) (Fig. 1, G–I). As a control, Kir2 inhibition with ML133 did not affect zebrafish chemotaxis (Fig. S1 B).

### Kir regulates the MP and chemotaxis of primary human neutrophils
We then validated our findings in primary human neutrophils (PMNs). Human peripheral blood neutrophils express 14 members of the Kir gene family, with high levels of *KCNJ2* and *KCNJ15*, and intermediate levels of *KCNJ1* and *KCNJ13* (Rincón et al., 2018). KCl, which shifts the K⁺ reversal potential and induces cell depolarization (Kilbourne et al., 1991; Rienecker et al., 2020), and BaCl₂, a general Kir blocker (Gilles, 2022), are also used. KCl, BaCl₂, and the relatively selective Kir inhibitors all reduced

neutrophil chemotaxis toward LTB4, significantly decreasing their directionality toward the chemokine (Fig. 1, J–L and Video 1). Collectively, these findings suggest that the Kir family is essential for effective gradient sensing during neutrophil chemotaxis in both humans and zebrafish. Mouse neutrophils mainly express *Kcnj8*, *10*, *5*, and *13*. VU590 did not influence the chemotaxis of mouse bone marrow neutrophils (Fig. S1, C–E), although BaCl₂ significantly inhibited their movement. These results demonstrate divergence in Kir expression and a dose-dependent function across evolution.

### Neutrophil intrinsic Kir7.1 regulates chemotaxis in zebrafish
For genetic inhibition, we used a dominant-negative Jaguar-like zebrafish (*Kir7.1*[G157E]) that showed an altered pigment pattern caused by a point mutation in the germline (Iwashita et al., 2006; Silic et al., 2020). The neutrophil response to LTB4 or tail wounds was consistently reduced in the *Kir7.1*[G157E] fish (Fig. 2, A–D). We bred the *Kir7.1*[G157E] line to visualize cell responses with a transgenic zebrafish line expressing mCherry in neutrophils. Neutrophils failed to migrate to the ventral fin with a specific defect in directness and directional migration, although cell speed remained normal (Fig. 2, E–H and Video 2). Conversely, neutrophil chemotaxis was intact in a *kcnj13* knockout fish line (*kcnj13*[pu109]) (Silic et al., 2020), possibly due to redundant functions of other Kir family members or genetic compensation (Fig. S1, F–H).

To specifically disrupt Kir7.1 function in neutrophils, we created two transgenic zebrafish lines using a neutrophil-specific lysozyme C (lyzC) promoter (Hsu et al., 2019): *Tg(lyzC:kcnj13-2A-mCherry)*[pu44] and *Tg(lyzC:kcnj13-Q153H-2A-mCherry)*[pu45]. These lines overexpress either WT or a dominant-negative version of Kir7.1, called Kir7.1[Q153H], along with mCherry via a cleavable 2A peptide (Fig. 2 I). Since Kir channels function as tetramers, their conductance decreases if the tetramers are composed of both WT and mutant proteins (Silic et al., 2020). Neutrophils from *Tg(lyzC:mCherry)*[pu43], which express only mCherry, serve as controls. Overexpression of WT or dominant-negative Kir7.1 reduced neutrophil recruitment to wounds, indicating that channel activity must be within an optimal range to support chemotaxis (Fig. 2, J and K). Specifically, Kir7.1[Q153H] overexpression significantly decreased neutrophil chemotaxis toward LTB4 without affecting their random migration (Fig. 2, L–O). Treating the Kir7.1[Q153H] line with VU590 did not further diminish chemotaxis, suggesting that VU590 primarily targets Kir7.1 in zebrafish neutrophils (Fig. S1 I).

### Kir7.1 regulates the spatial gradient of plasma MP during neutrophil chemotaxis
To explore how MP modulates neutrophil chemotaxis, we used the genetically encoded voltage indicator ASAP3 (accelerated sensor of action potentials), which has been employed to monitor action potentials in neurons (Villette et al., 2019). The ASAP family sensors are well established and calibrated in cell culture, and they decrease brightness upon depolarization. We created a transgenic line, *Tg(lyzC:ASAP3)*[pu40], to express ASAP3 specifically in zebrafish neutrophils. This line was bred with *Tg(lyzC:mCherry-CAAX)*[pu41] to label the cell membrane, thereby reducing

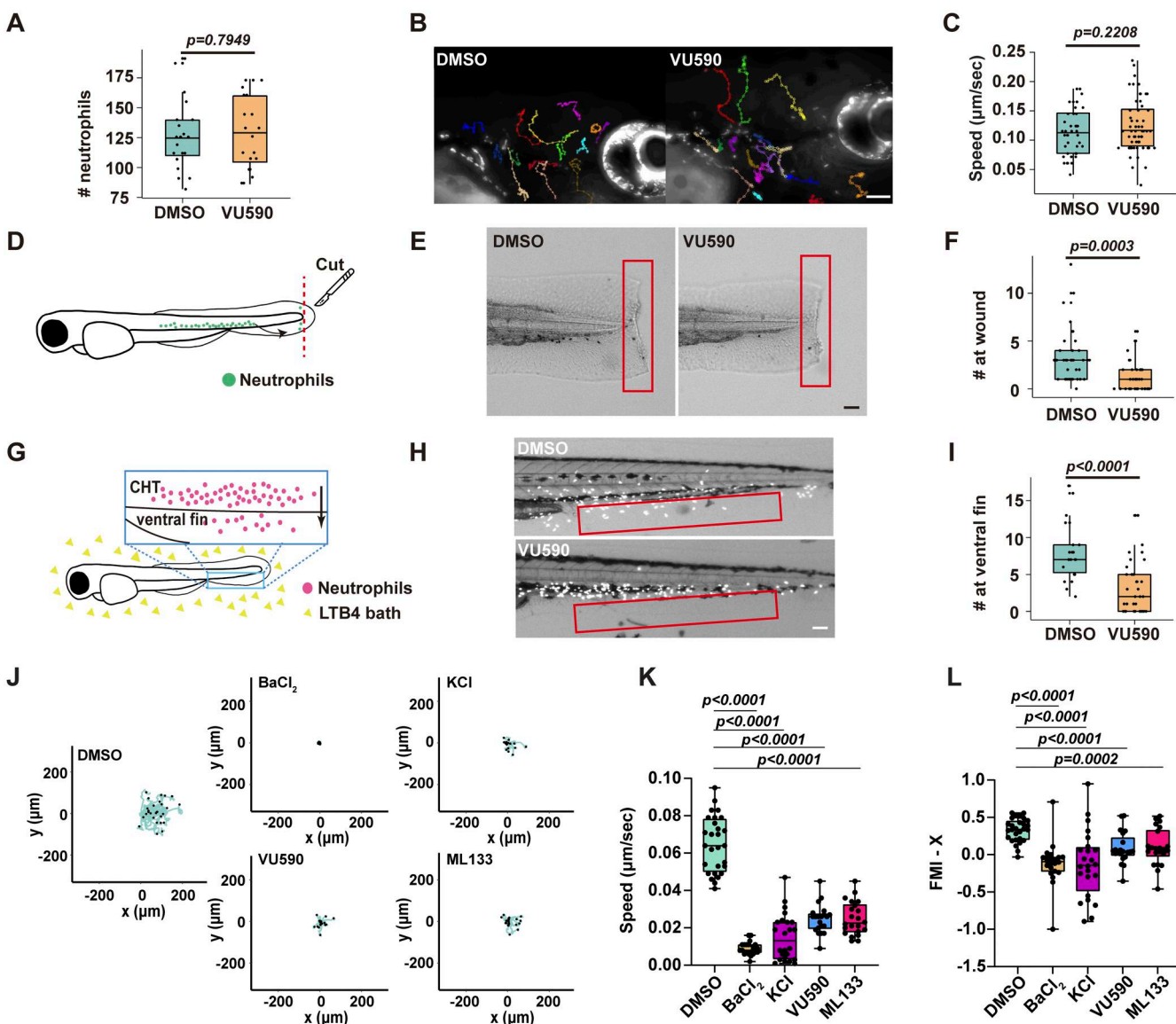

Figure 1. **Pharmacological inhibition of Kir7.1 attenuates neutrophil chemotaxis in zebrafish. (A)** Number of total neutrophils in the zebrafish 3 dpf embryos treated with 50 μM VU590 or DMSO. Each dot represents one fish. *n* > 20 in each group. Data representative of three independent experiments. **(B and C)** Representative tracks and (C) speed of zebrafish neutrophil spontaneous migration with VU590 or DMSO treatment using *Tg(lyzC:mCherry)pu43*. Scale bar: 50 μm. Each track represents one neutrophil. *n* > 15 neutrophils from 3 fish of each group. **(D)** Schematic illustration of tail transection and neutrophil recruitment. Green dots represent neutrophils. Neutrophils migrating from CHT to the tail 30 min after injury were counted. **(E and F)** Representative images and (F) quantification of neutrophils recruited to the ventral fin in embryos with VU590 or DMSO treatment. Neutrophils in the boxed regions are quantified. Scale bar: 50 μm. *n* > 20 in each group. Data representative of three independent experiments. **(G)** Schematic illustration of the LTB4-induced neutrophil chemotaxis assay. Red dots represent neutrophils. Neutrophils migrate from CHT to the ventral fin fold in the LTB4 bath. 30 min after injury were counted. **(H and I)** Representative images and quantification (I) of neutrophils migrating to the ventral fin fold 30 min in an LTB4 bath with VU590 or DMSO treatment. *n* > 20 in each group. Data representative of three independent experiments. Scale bar: 50 μm. (A, C, F, and I) Results are presented as mean ± SD, Mann–Whitney test. **(J)** Representative tracks of human PMN chemotaxis toward LTB4 after treatment with DMSO, BaCl₂, KCl, VU590, or ML133 in an under-agarose migration plate coated with collagen IV. The lines represent individual cell trajectories over a 60-min period. **(K and L)** Quantification of speed and forward migration index (FMI). *n* > 15 in each group. Data present mean ± SD, representative of three independent experiments. Multiple comparisons and one-way ANOVA.

potential artifacts from changes in membrane thickness, which can affect local ASAP3 sensor levels. We first imaged the same neutrophils in the CHT before and after VU590 treatment and observed a specific decrease in ASAP3 intensity, indicating depolarization (Fig. S2 A). During chemotaxis toward LTB4, neutrophils showed a spatial MP gradient along their axis of movement (Fig. 3, A and B, and Video 3). Cell velocity was cyclic,

with correlated protrusion and tail retraction phases (Fig. 3, C–E). A higher MP ratio (relatively hyperpolarized) was linked to retraction speed (Fig. 3 F). The protrusion was relatively depolarized, and slight hyperpolarization correlated with reduced protrusion speed (Fig. 3 G). This MP gradient was cyclic, with the most prominent changes occurring during active protrusion (Fig. 3, H–J). Conversely, this gradient was not observed during

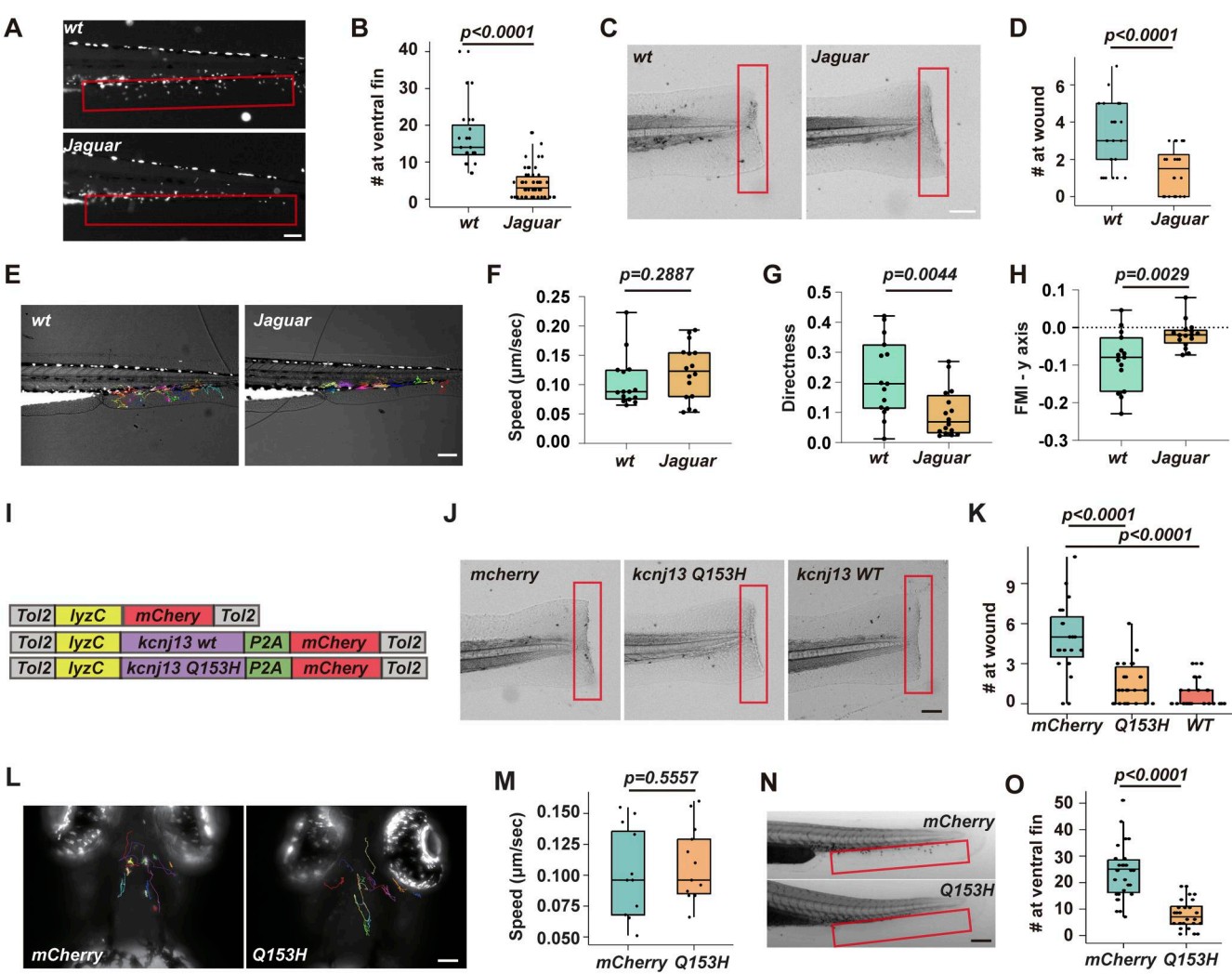

**Figure 2.** **Kir7.1 regulates zebrafish neutrophil chemotaxis. (A and B)** Representative images and quantification (B) for neutrophil response to LTB4 in the *Jaguar* or WT sibling controls. Scale bar: 50 μm. *n* > 20 in each group. Data representative of three independent experiments. **(C and D)** Representative images and quantification (D) for neutrophil response to tail wounding in the *Jaguar* or WT sibling controls. Scale bar: 50 μm. *n* > 20 in each group. Data representative of three independent experiments. **(E)** Representative tracks of neutrophil migration to the ventral fin in response to LTB4 in the *Jaguar* or WT sibling controls. Each track represents one neutrophil. Scale bar: 50 μm. **(F–H)** Quantification of the speed (F), directness (G), and forward migration index (H). *n* >20 in each group. Data representative of three independent experiments. **(I)** Construct design for neutrophil-specific kcnj13 WT and Q153H mutation overexpression in zebrafish to generate *Tg(lyzC:kcnj13-2A-mCherry)*[pu44] and *Tg(lyzC:kcnj13-Q153H-2A-mCherry)*[pu45]. **(J and K)** Representative images and (K) quantifications of neutrophil recruitment to the tail fin wound. Scale bar, 300 μm. *n* > 20 in each group. Data representative of three independent experiments. **(L and M)** Representative tracks and (M) speed of neutrophils' spontaneous migration in the head mesenchyme in transgenic lines with neutrophil-specific Kir7.1 Q153H overexpression; *Tg(lyzC:mCherry)*[pu35]-expressing mCherry alone is used as a control. Scale bar: 50 μm. *n* > 15 neutrophils from 3 fish of each group. **(N and O)** Representative images and (O) quantification of the number of neutrophils recruited to the ventral fin upon LTB4 treatment in the indicated zebrafish transgenic lines overexpressing mCherry or Kir7.1Q153H. Scale bar: 50 μm. *n* > 20 in each group. Data representative of three independent experiments. (B, D, F–H, M, and O) Results are presented as mean ± SD, Mann–Whitney test. (K) Multiple comparisons and Kruskal–Wallis test.

neutrophil random migration in the head, indicating a response specifically triggered by chemokines (Fig. S2 B and Video 4). Additionally, we expressed Kir7.1-GFP fusion in zebrafish neutrophils and observed asymmetric localization during chemotaxis but not during random migration (Fig. S2 C). The dynamic heterogeneity of MP in neutrophils was unexpected, although we cannot rule out all possible artifacts with biosensor imaging. Previous studies mainly focused on the overall MP of the cell without resolving its spatiotemporal variations. Given the rapid speed at which electrical signals propagate within the cell (within sub milliseconds), any transient heterogeneity in action

potential distribution was considered too brief to detect. Indeed, the poroelastic structure of the actin meshwork and cytoplasm within the lamellipodium resists water flow, possibly preventing or slowing the immediate equalization of local ion concentrations (Charras et al., 2005; Ito et al., 1992). Moreover, it has been suggested that mitochondrial MPs can be highly heterogeneous. Individual cristae within the same mitochondrion can have different MP (Smiley et al., 1991), potentially functioning as independent bioenergetic units (Wolf et al., 2019).

To assess the effect of Kir7.1 on neutrophil plasma MP, we created two additional zebrafish lines, *Tg(lyzC:kcnj13-2A-mCherry-*

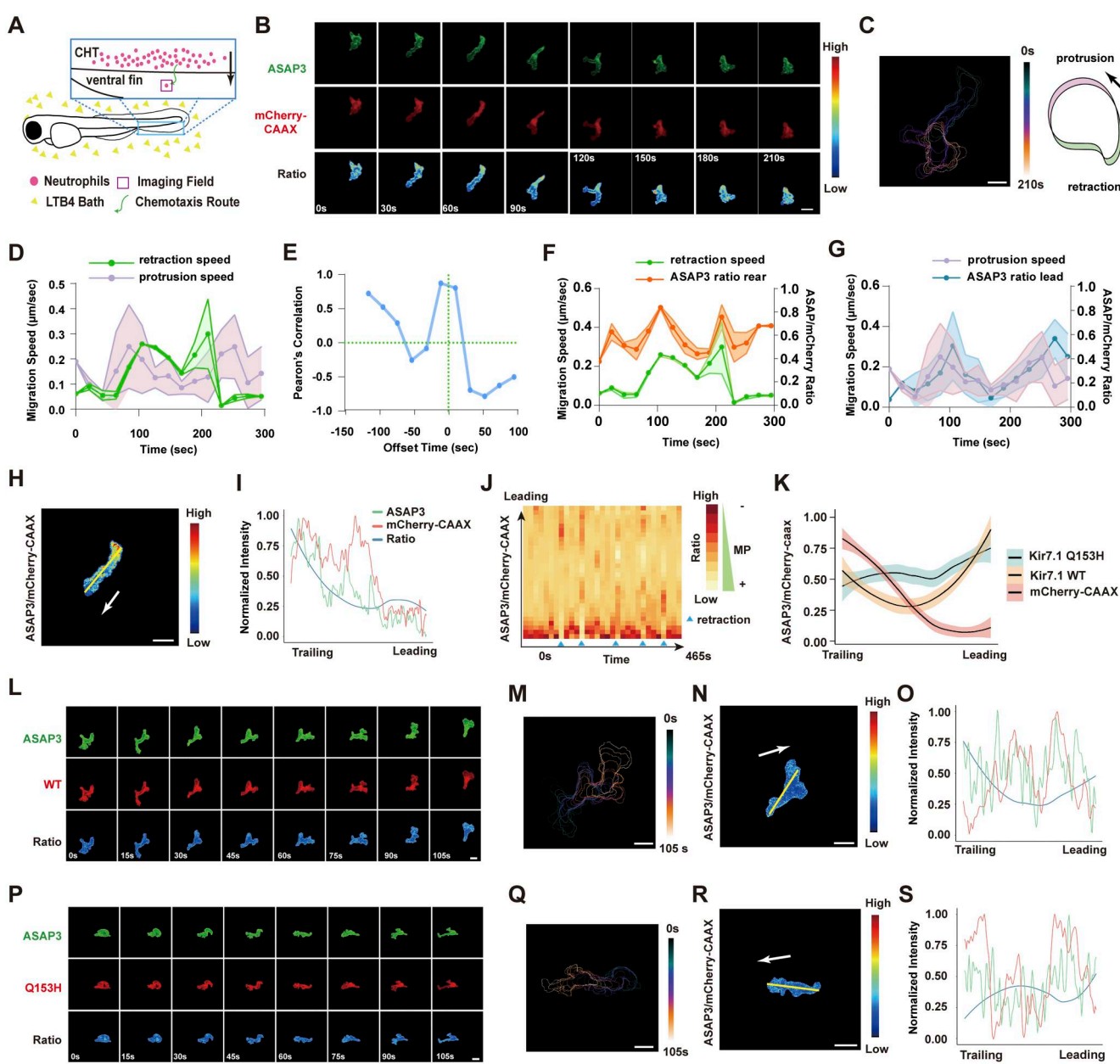

Figure 3. **Kir7.1 regulates plasma MP gradient during neutrophil chemotaxis. (A)** Schematics of the neutrophils being imaged in the zebrafish ventral fin fold when migrating toward an LTB4 gradient. **(B)** Representative time-lapse imaging of ASAP3 and mCherry-CAAX in fish neutrophils migrating toward 30 nM LTB4 using *Tg(lyzC: ASAP3;lyzC: mCherry-CAAX)*. Ratiometric analysis (ASAP3/mCherry-CAAX) was performed. **(C)** Cell outlines over time, indicating changes in cell shape during migration. Time was color-coded. Scale bar, 10 μm. Cell outlines were used to quantify the speeds of cell protrusion (green) and retraction (magenta). **(D)** Protrusion (green) and retraction (magenta) speeds for the cell shown in B. Data are presented as mean ± SD for *n* = 3 spots in each group. **(E)** Pearson's correlation between the protrusion and retraction speeds for the cell in B with the indicated time offset. **(F and G)** ASAP3 ratio and the correlated migration speed at the retraction and the protrusion of the cell are shown in B. Data are presented as mean ± SD for *n* = 3 spots in each group. **(H)** A representative frame of the cell in B and the ratio of ASAP3/mCherry-CAAX are color-coded. Scale bar, 10 μm. **(I)** Quantification of fluorescence intensity along the indicated line in H. green: ASAP3; red: mCherry-CAAX; blue: ASAP3/mCherry-CAAX ratio. **(J)** Kymograph plotting the dynamics of the MP gradient change. The time interval between frames is 15 s. Blue triangles indicate the tail retraction phase during neutrophil migration. The ratio indicates ASAP3/mCherry-CAAX. **(K)** Quantitative analysis of the normalized ratio of ASAP3/mCherry along the axis of neutrophil migration upon Kir7.1 WT or Q153H overexpression compared with mCherry-CAAX control. Data are presented as mean ± SD for *n* = 15 cells in each group. **(L–S)** similar analysis of (B, C, H, and I) using *Tg(lyzC: ASAP3;lyzC: kcnj13-2A-mCherry-CAAX)* or *Tg(lyzC: ASAP3;lyzC:kcnj13-Q153H-2A-mCherry-CAAX)*.

CAAX)^pu46 and *Tg(lyzC:kcnj13-Q153H-2A-mCherry-CAAX)^pu47*, which overexpress Kir7.1 WT or Q153H fused with mCherry-CAAX via the P2A peptide. Ratiometric imaging of ASAP3/mCherry-CAAX in neutrophils showed a significant reduction in the MP gradient with overexpression of either construct (Fig. 3, K–S and Video 5). However, overall cell polarity remained intact, as indicated by proper localization of PI3K products, phosphatidylinositol (4,5)-bisphosphate, and phosphatidylinositol (3,4,5)-trisphosphate at

the leading edge during migration (Yoo et al., 2010), though not directed toward the chemokine source (Fig. S2 D). Likewise, the intracellular calcium gradient was unaffected (Fig. S2 E), possibly due to the rapid diffusion of calcium in the cytoplasm. Alternatively, GPCRs release calcium from the ER—which is not regulated by the MP—while the MP can control $Ca^{2+}$ entry at the plasma membrane through store-operated calcium entry (Kozak and Putney, 2017). These findings show that Kir7.1 influences the MP gradient during neutrophil chemotaxis, which is probably vital for gradient sensing without impacting neutrophil development, random migration, or cell polarization.

## Plasma membrane depolarization induces protrusions and guides neutrophil migration

We then used optogenetic actuators, validated in zebrafish, to study how plasma MP influences neutrophil migration (Antinucci et al., 2020). CoCHR produces high-amplitude photocurrents, allowing cations, mainly $Na^+$, to enter, which depolarizes cells (Antinucci et al., 2020). We created a transgenic line $Tg(lyzC:CoChR-mCherry)^{pu42}$. To verify that CoCHR can depolarize neutrophils enough, we illuminated neutrophils co-expressing ASAP3 and CoChR. The ASAP3 signal decreased across the cells after focal photo-stimulation, likely due to rapid ion spread (CoCHR deactivation time is 20 ms, much shorter than the time needed for the system to switch from photo-manipulation to acquisition mode [Antinucci et al., 2020]) (Fig. 4, A and B, and Video 6). Cells quickly recovered from the MP and kept migrating. Next, we tested whether local depolarization of the cell membrane could affect directional migration or start new neutrophil protrusions. During tissue migration, neutrophils form multiple protrusions, with one becoming dominant. We randomly chose one protrusion for stimulation, and 8 out of 10 became dominant and directed cell migration (Fig. 4 C and Video 7). While actively migrating, we targeted one side of the cell body, about one-third from the front. Small transient protrusions formed, lasting 30 s before retracting into the cell body in 7 out of 10 cells (Fig. 4 C and Video 8). Stimulating the tail did not slow neutrophil movement (Fig. 4 C and Video 9). This finding aligns with previous optogenetics studies, which have shown that cell polarity remains unchanged during persistent migration (Yoo et al., 2010). To test if GPCR activation is needed for depolarization-driven cell polarization, we expressed the catalytic domain of pertussis toxin in zebrafish neutrophils (Hammerschmidt and McMahon, 1998). These neutrophils did not respond to LTB4 (Fig. S3 A). Local depolarization-induced protrusions also failed to form (Fig. S3 B), indicating that depolarization requires Gα signaling to induce cell polarization.

To further understand the role of MP, we employed additional optogenetic tools. GtACR2, an anion-selective channel, and BLINK2, a $K^+$-selective channel, induce hyperpolarization and suppress neuronal activity (Fig. S3, C and D). We delivered the actuators into zebrafish neutrophils through transient plasmid injection and exposed the entire cell to blue light (Fig. S3, E and F; and Video 10). Cell migration stopped immediately after light stimulation but recovered within 2–3 min after the stimulus was stopped. Control cells expressing mCherry were

unaffected by similar light stimulation conditions. Our findings suggest that neutrophil behavior is excitable, much like the behavior of neurons.

## Kir regulates GPCR signaling in human neutrophil-like cells

Due to the limited number of cells and their inaccessibility within the tissue, we next turned to human neutrophil cell models, specifically differentiated HL-60 cells (dHL-60). We performed an under-agarose migration assay on BSA or collagen-coated surfaces and multiple chemoattractants, including fMLP, LTB4, and the complement fragment C5a, to determine if Kir activity is required for general chemotaxis. VU590 reduced neutrophil directionality toward the gradient in all conditions tested (Fig. 5, A and B). Notably, the cell speed was not significantly different from the DMSO control (Fig. 5 C). As a control, we measured cell random migration under agarose. VU590 treatment increased cell speed considerably; however, in the presence of fMLP (chemokinesis), the speed was comparable with the DMSO-treated control. We attempted to express KCNJ13-DN in dHL-60 cells but did not observe a robust phenotype, which is likely due to the high expression of other KCNJ family members, such as KCNJ2 and KCNJ15 (Rincón et al., 2018).

We then measured the MP using a whole-cell patch clamp. dHL-60 cells had a resting MP of about –30 mV, and fMLP caused depolarization. VU590 and KCl depolarized dHL-60 cells to 0 mV, which did not change further after fMLP stimulation (Fig. 5 E). These data suggest that the Kir family helps maintain the polarized MP in neutrophil-like cells at rest. Chemokine binding to chemokine receptors, GPCRs, triggers rapid activation of downstream signaling pathways, including calcium mobilization from intracellular stores and p21-activated kinase (PAK) (Itakura et al., 2013), which localizes to the leading edge of migrating neutrophils and promotes effective directional migration (Fig. 5 F). Both the calcium spike and PAK activation caused by fMLP stimulation were reduced in UV590-treated dHL-60 cells (Fig. 5, G–I). KCl also significantly suppressed PAK activation, indicating that depolarization is necessary for optimal GPCR signaling. Therefore, Kir7.1 may regulate neutrophil directional sensing by setting the resting MP to maintain excitability, reinforcing local depolarization at the protrusion, or inducing repolarization to allow cycles of pseudopod selection, similar to Kir's role in cardiac myocytes (Hibino et al., 2010). It is unlikely that Kir7.1 controls cell volume, which is related to overall cell speed (Nagy et al., 2024, Preprint).

Whereas Dictyostelium discoideum cells (Dicty) are a well-established model for chemotaxis, the MP did not affect their motility or chemotaxis to cAMP (Gao et al., 2011; Van Duijn et al., 1990). This discrepancy may be due to differences in GPCR signaling in Dicty and neutrophils. In Dicty, cAMP activates Gα2βγ, cAMP/cGMP production, whereas in neutrophils, cAMP is not induced. In addition, Vm in Dicty is maintained by an electrogenic proton pump, unlike $K^+/Na^+$ pumps in neutrophils. Most importantly, in systems where Vm is required for chemotaxis, such as neutrophils and Physarum polycephalum (Ueda et al., 1975), chemotactic factors induce ion fluxes and changes in

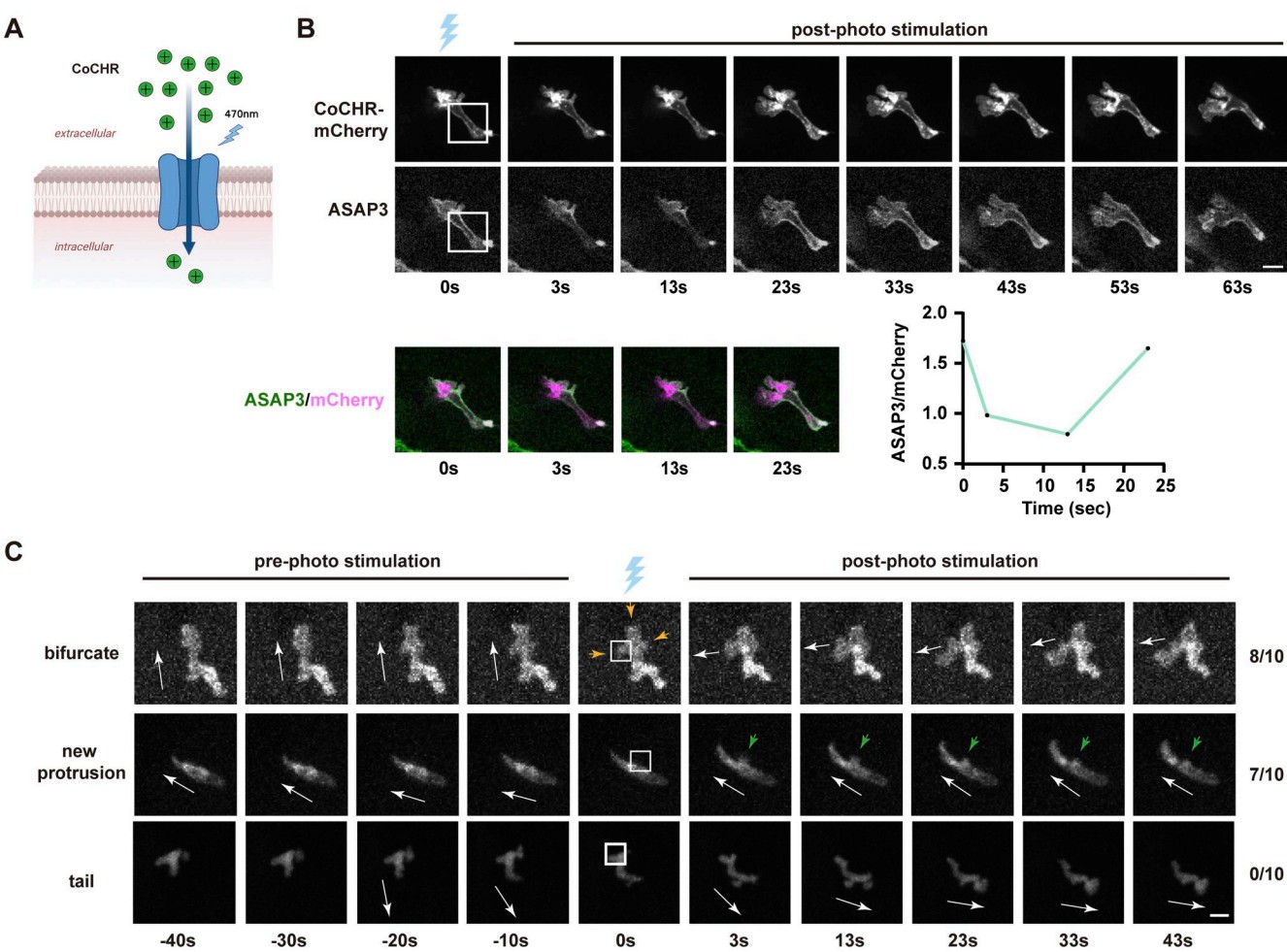

Figure 4. **Plasma membrane depolarization induces protrusions and guides the migration of neutrophils. (A)** Schematic of the ion selectivity of CoChR. **(B)** Time-lapse of ASAP3 and mCherry after photo stimulation using *Tg(lyzC: ASAP3;lyzC: CoChR-mCherry)^pu42*. The ASAP/mCherry ratio is calculated to reflect the transient depolarization. **(C)** Representative time-lapse images of neutrophil movement before and after the laser stimulation on the indicated subcellular areas using *Tg(lyzC: CoChR-mCherry)^pu42*. The blue laser cartoon indicates the photo-stimulation frame. White arrowheads indicate the direction of cell movement. White boxes indicate the area stimulated. Orange arrowheads represent multiple protrusions. Green arrowheads represent the de novo protrusion. Scale bar, 5 μm. *N* = 10 cells for each condition.

MPs. It is to be determined whether the chemotactic factor induces depolarization in Dicty cells. We speculate that GPCR-induced depolarization correlates with the functional importance of Vm in chemotaxis.

Related to chemotaxis, the electric field is expected to change the MP, which is necessary for galvanotaxis. Knocking down *KCNJ15* (encoding Kir4.2) in epithelial cells specifically eliminates directionality, not speed, during galvanotaxis, without impacting basal motility or directional migration in a monolayer scratch assay (Nakajima et al., 2015). Therefore, Kir channels may regulate the MP during both galvanotaxis and chemotaxis, although the specific channels and biological processes involved vary depending on the cell type. While Vm mainly influences membrane protein structure and activity, electric fields also bias directionality by attracting charged molecules (Belliveau et al., 2024, *Preprint*). Another related study shows that inner membrane charge also influences chemotaxis in both Dicty and neutrophil-like cells (Banerjee et al., 2022). The mechanism here is likely based on charge interactions.

## Materials and methods

### Generation of transgenic and mutant zebrafish lines

The zebrafish experiment was conducted by internationally accepted standards. The Animal Care and Use Protocol was approved by the Purdue Animal Care and Use Committee (PACUC), in accordance with the guidelines for using zebrafish in the NIH Intramural Research Program (protocol number: 1401001018). Microinjections of fish embryos were performed by injecting 1 nl of a mixture containing 25 ng/μl plasmids and 35 ng/μl tol2 transposase mRNA in an isotonic solution into the cytoplasm of embryos at the one-cell stage. The stable lines were generated as previously described (Deng et al., 2011). Each founder candidates were outcrossed with WT, and offspring carrying the reporter genes were raised. At least two founders (F0) were obtained for each line. Experiments were performed using F2 larvae produced by F1 fish derived from multiple founders to minimize the artifacts associated with random insertion sites.

*The Kcnj13* knockout line was published (Silic et al., 2020), generated using gRNAs targeting the 6th exon of the *kcnj13* gene.

Figure 5. **Kir7.1 regulates chemotaxis and GPCR signaling of human neutrophils. (A)** Under-agarose migration assay of dHL-60 cells treated with 50 μM VU590 or DMSO on BSA- or collagen-coated glass-bottom dishes toward indicated chemoattractants. **(B and C)** Quantification of forward migration index (FMI) toward the chemokine gradient and velocity of cells. $n > 10$ in each group. Data present mean ± SD, representative of three independent experiments. **(D)** Under-agarose random migration velocity of dHL-60 cells treated with 50 μM VU590 or DMSO on BSA-coated glass-bottom dishes in the presence or absence of uniform fMLP. $n = 20$ in each group. Data present mean ± SD, pooled from three independent experiments. **(E)** Whole-cell patch-clamp measurement of dHL-60 cells treated with VU590 or KCl with or without fMLP activation. $n > 10$ in each group. Data present mean ± SD, representative of three independent experiments. **(F)** Illustration of how Kir regulates MP and GPCR signaling. **(G)** Quantification of intracellular calcium after fMLP treatment of dHL-60 cells treated with VU590 or DMSO. Data present mean ± SD, representative of three independent experiments. **(H and I)** Immunoblot and normalized quantification of time-course PAK activation after fMLP treatment of dHL-60 cells treated with DMSO, VU590, or KCl. Data present mean ± SD from three independent experiments. **(B, C, and G)** Mann–Whitney test. **(D, E, and I)** Multiple comparisons and one-way ANOVA. Source data are available for this figure: SourceData F5.

The mutation was confirmed by Sanger sequencing. Before being used for experiments, the mutant fish were outcrossed to the F3 generation.

## Molecular cloning

A tol2-lyzC vector was used in this study for gene expression in fish neutrophils. Target genes were inserted into the Tol2 backbone, which contains the lyzC promoter and SV40 polyA, for neutrophil-specific expression. Fish *kcnj13* WT and Q153H mutation gene fragments were cloned from the constructs described in our previous studies (Silic et al., 2020). ASAP3 gene was cloned from pAAV-hSyn-ASAP3-WPRE plasmid (#132331; Addgene); GtACR2 gene was cloned from pTol1-UAS:GtACR2-tdTomato plasmid (#124236; Addgene); BLINK2 gene was cloned from pDONR-BLINK2 plasmid (#117075; Addgene); CoCHR gene was cloned from pAAV-Syn-CoCHR-GFP plasmid (#59070;

Addgene); PHAKT fragment was cloned from pHR SFFVp PHAkt Cerulean plasmid (#50837; Addgene) and fused to EGFP. GFP-P2A-PTX was cloned from pJR019 (#198834; Addgene). The mCherry and mCherry-CAAX fragments used in fish and human cells were cloned from our previous plasmid constructs in the lab and inserted into fish and human vectors, respectively.

## Pharmacological treatment

VU590 (item no. 15177; Cayman Chemical) or ML133 (item no. 31232; Cayman Chemical) was dissolved in DMSO to make a 100 mM stock and then further diluted in E3 or mHBSS (modified HBSS with 20 mM HEPES and 0.5% FBS) to a working concentration of 50 μM. DMSO was utilized as a vehicle control. For neutrophil recruitment and random motility assays, larvae were pretreated with the inhibitor for 1 h before experimental procedures. BaCl$_2$ and KCl were prepared at a 2 M stock

concentration and diluted to a working concentration of 40 mM for 1 h pretreatment on cells. Working concentrations were maintained throughout the experiment.

## PMN isolation

PMNs were obtained from the peripheral blood of healthy adult donors and collected under approval by Purdue University's Institutional Review Board. PMNs were isolated from 20 ml of blood with the Miltenyi MACSxpress Neutrophil Isolation Kit (#130-104-434; Macs) and RBCs lysed for 10 min at RT (#00-4300-54; Thermo Fisher Scientific) as described (Hsu et al., 2022) according to the manufacturer's instructions.

## Primary mouse neutrophil isolation

Mice were anesthetized with isoflurane and euthanized by cervical dislocation, per protocols approved by the Institutional Animal Care and Use Committee. The femurs and tibias were harvested, and all surrounding skin, muscle, and tendons were carefully removed using sterile tools and clean paper towels. Cleaned bones from each mouse were placed into individual wells of a 6-well plate containing ice-cold bone collection buffer (DPBS supplemented with penicillin/streptomycin). Following the dissection of all bones, both epiphyses were cut off using sterile scissors. Each bone was then placed cut-side down into a 0.6-ml microcentrifuge tube that had a small hole punctured at the bottom using an 18-G sterile needle. The 0.6-ml tube was nested inside a 1.5-ml microcentrifuge tube. Bone marrow was extracted by centrifugation at $10,000 \times g$ for 30 s at RT. After centrifugation, the bones were discarded, and the collected marrow was resuspended in 1 ml of cold bone collection buffer, with care taken to minimize the formation of air bubbles to prevent neutrophil activation. The cell suspension from each sample was gently transferred onto a 40-μm cell strainer placed over a 50-ml conical tube, and all samples were pooled together. The strainer was rinsed with an additional 10 ml of bone collection buffer to maximize cell recovery. The filtered bone marrow cells were kept on ice for 30 min before further processing. Neutrophils were then isolated by negative selection using the Miltenyi Mouse Neutrophil Isolation Kit (Catalog #130-097-658), following the manufacturer's magnetic bead–based protocol.

## Under-agarose migration assay for primary neutrophils

The under-agarose chemotaxis assay used in this study was modified from a previous study (Heit and Kubes, 2003). Chemoattractant reservoirs were prepared using 8-well Collagen IV–coated IBIDI μ-Slides (cat. no: 80822; IBIDI). A 3% low-melting-point agarose solution was prepared in DPBS (without calcium and magnesium), cooled to 37°C, and mixed 1:1 with pre-warmed mHBSS medium (HBSS without calcium and magnesium, supplemented with 1% heat-inactivated FBS and 20 mM HEPES). Human or mouse neutrophils were suspended at $3 \times 10^6$ cells/ml, and 5 μl of the cell suspension was seeded at the center of each well. Cells were allowed to settle at RT for 10 min. Subsequently, 650 μl of the agarose/media mixture was gently overlaid onto each well without disturbing the cells and left to solidify at RT for 20 min. A small chemoattractant reservoir

(~2 mm width) was created by cutting the agarose near the right edge of each well using a sterile surgical blade. The slides were then incubated at 37°C for 30 min before imaging. Just prior to imaging, 400 nM LTB4 in mHBSS was added to the reservoir. Time-lapse live imaging was initiated immediately after the warming step, with image acquisition every 1 min for 1 h. Live-cell imaging was performed using a Nikon Ti Eclipse spinning disk confocal microscope equipped with a Yokogawa CSU-W1 module and a 10 × phase DIC objective. Cell tracking was performed using the MTrackJ plugin in ImageJ.

## Under-agarose chemotaxis assay for dHL-60 cells

The under-agarose chemotaxis assay applied in this study was modified from that in a previous study (Heit and Kubes, 2003). Briefly, 0.5% ultrapure agarose gel made from 1% gel in DPBS 1:1 ratio to warm mHBSS was poured into a 35-mm collagen-coated tissue culture petri dish. 3 ml gel was required for each dish. The gel was left at 4°C for 15 min to solidify. Wells were punched by a 1.5-mm diameter metal tube with 0.3 mm wall thickness using a customized 3D printing mold. The distance between the two wells designed in the mold is 2 mm. $10^6$–$10^7$ pretreated HL-60 cells in mHBSS were loaded into one well and incubated at 37°C for 15 min 500 nM fMLP, 600 nM LTB4, or 10 μg/ml C5a in mHBSS were loaded into the adjacent well until the liquid level reached the top. The petri dish was incubated at 37°C for 15 min, then set up in BioTek Lionheart FX Automated Microscope using a 10× phase objective, Plan Fluorite WD 10 NA 0.3 (1320516) for imaging. Cells were tracked using MTrackJ image J plugin.

## Immunoblotting

dHL-60 cells were differentiated for 6 days in 1.3% DMSO, and $10^6$ cells were pelleted in microcentrifuge tubes. Cells were lysed with RIPA buffer (#89900; Thermo Fisher Scientific), supplemented with a proteinase inhibitor cocktail (#4693132001; Roche). Protein contents in the supernatant were determined with the BCA assay (#23225; Thermo Fisher Scientific), and 25–50 μg of proteins was subjected to SDS-PAGE and transferred to PVDF membranes. Phosho-protein probing was performed as described (Graziano et al., 2017). Briefly, $1.5 \times 10^6$ cells/ml dHL-60s were starved in serum-free media for 1 h at 37°C/5% $CO_2$. Cells were then stimulated with 10 nM fMLP for 1 or 3 min at RT and immediately put on ice with the addition of ice-cold lysis solution in a 1:1 ratio (20% TCA, 40 mM NaF, and 20 mM β-glycerophosphate [#T6399, #G9422, #s6776; Sigma-Aldrich]). Cells were incubated on ice for 1 h, and proteins were pelleted down and washed once with 0.75 ml of ice-cold 0.5% TCA and resuspended in 2× Laemmli sample buffer (Bio-Rad Laboratories). Western blot was performed with anti–phospho-PAK1/2 (#2605S; Cell signaling), anti-PAK (#2604; Cell signaling), and vinculin as a loading control. Membranes were incubated with primary antibodies O/N at 4°C, then with goat anti-mouse (35518; Invitrogen) or anti-rabbit (SA5-35571; Thermo Fisher Scientific) secondary antibodies. The fluorescence intensity was measured using an Odyssey imaging system (LI-COR). Band intensity was quantified using Image Studio (LI-COR) and normalized with respective loading controls.

## Patch clamp of dHL-60 cells

The neutrophils were centrifuged at 1,000 × g for 5 min in 24-well plates with a surface-treated polystyrene-coated glass slide in each well to make the neutrophils loosely adherent to the glass slide. Filamented borosilicate glass capillaries (BF150-86-10, Sutter Instruments) were pulled by a micropipette puller (P-1000, Sutter Instruments) for whole-cell patch-clamp recording electrodes to a resistance of 4–6 MΩ. The glass electrodes were filled with an internal solution containing 20 mM KCl, 100 mM K-gluconate, 10 mM HEPES, 4 mM MgATP, 0.3 mM Na2GTP, and 7 mM phosphocreatine. The pH was adjusted to 7.4, and the osmolarity was adjusted to 300 mOsm. The cultured neutrophils on the glass slide were transferred to the recording chamber perfused with a modified Tyrode bath solution (150 mM NaCl, 4 mM KCl, 2 mM MgCl$_2$, 10 mM glucose, 10 mM HEPES, and 2 mM CaCl$_2$ [pH 7.4, 310 mOsm]) (Chubykin et al., 2007) at 32°C. fMLP, VU590, and KCl were added to the Tyrode bath accordingly. The whole-cell patching was conducted using open-source software, Autopatcher IG, developed previously (Wu and Chubykin, 2017; Wu 吴秋雨 et al., 2016). The patch-clamp recording signals were processed with an amplifier (Multiclamp 700B, Molecular Devices) and a digitizer (Digitata, 1550, Molecular Devices). Acquired data were collected using Clampex software (Axon Instruments) with a 10,000-Hz low-pass filter. Continuous potential recordings by the current clamp were filtered at 2 kHz. The resting MP was calculated by averaging 0–1 s continuous potential measurement.

## RT-PCR

Total RNA of neutrophils sorted from adult kidney marrow (Hsu et al., 2019) was extracted using an RNeasy Plus Mini Kit (no. 74104; Qiagen). RT-PCR was performed with Qiagen OneStep RT-PCR Kit (210212). The following primers were used: *dre-ef1a+*: 5′-TACGCCTGGGTGTTGGACAAA-3′; *dre-ef1a−*:5′-TCTTCTTGA TGTATCCGCTGA-3′; *dre-mpx+*: 5′-ACCAGTGAGCCTGAGACAC GCA-3′; *dre-mpx−*: 5′-TGCAGACACCGCTGGCAGTT-3′; *dre-krt4+*: 5′-CTATGGAAGTGGTCTTGGTGGAGG-3′; *dre-krt4−*: 5′-CCTGAAGAGCATCAACCTTGGC-3′; *dre-kcnj1a1+*: 5′-CTCCA CACTGAAGAAAGAGCTTGCT-3′; *dre-kcnj1a1−*: 5′-ATTTTGCC ACAACAGGTCGCC-3′; *dre-kcnj1b+*: 5′-GGTTACGTGTTTCTC CCCGT-3′;

*dre-kcnj1b-*: 5′-ACAAGCAGCTCGCTAGGTTT-3′; *dre-kcnj2b+*: 5′-ATCGGCTTGAGCTTTGACCAAC-3′; *dre-kcnj2b−*: 5′-CTTTTC CATTGCCGTAGCCGT-3′; *dre-kcnj10a+*: 5′-TCGCAGACTAAAGAG GGGGA-3′; *dre-kcnj10a−*: 5′-TCCATCAGAGCCCCAGGTTA-3′; *dre-kcnj11+*: 5′-TTCGTCGCCAAAAACGGAAC-3′; *dre-kcnj11−*: 5′-GTGGACGGTGGCACTGATAA-3′; *dre-kcnj13+*:5′-AGGCCTTCATC ACTGGTGC-3′; *dre-kcnj13−*: 5′-TGGATCTCGTCAGGCAGGTA-3′.

## Tailfin wounding and Sudan black staining

Tailfin wounding and Sudan black staining were performed on embryos at 3 dpf, as previously described (Deng et al., 2011). Larvae were anesthetized in E3 supplemented with 0.1 mg/ml tricaine, and their tails were transected with a surgical blade. One hour after wounding, larvae were fixed in 1% formaldehyde in phosphate-buffered saline overnight at 4°C. The next morning, larvae were washed three times for 5 min each with PBS and stained with Sudan black (0.024% in 70% EtOH with 0.1% pheno) for 30 min, then washed with 70% EtOH for 5 min. The embryos were rehydrated with PBS plus 0.1% Tween 20 for 5 min, and pigments were cleared with 1% KOH/1% H$_2$O$_2$ for 10–15 min at RT. Finally, the larvae were transferred to PBS, and neutrophils were counted under a dissecting scope.

## Fluorescent confocal microscopy

Fluorescent microscopy imaging data were obtained with a Nikon A1R fluorescent confocal microscope. Larvae at 3 dpf were settled on a glass-bottom dish, and imaging was performed at 28°C. Neutrophil random motility imaging was described before (Deng et al., 2011).30 nM LTB4 was introduced to fish in E3 to induce neutrophil chemotaxis (Yoo et al., 2011). Time-lapse images for zebrafish neutrophil random migration and LTB4-induced chemotaxis were acquired using a 20× objective lens at 1-min intervals for 30 min. Biosensor imaging was acquired by a 60× oil objective lens. The green and red channels were acquired simultaneously with a 488- and 561-nm laser, respectively. Zebrafish neutrophils with biosensors were scanned and imaged by the z-stack function at 15-s intervals for 5 min.

## Optogenetics manipulation

Spinning-disk confocal microscopy (SDCM) was performed using a Yokogawa scanner unit CSU-X1-A1 mounted on an Olympus IX-83 microscope equipped with a 40 × 1.0–NA UPlanApo oil immersion objective (Olympus) and an Andor iXon Ultra 897BV EMCCD camera (Andor Technology). mCherry fluorescence was excited with the 561-nm laser line, and emission was collected through a 610/37-nm filter. Time-lapse images were collected at 10-s intervals to track the individual neutrophils, and z-series at 2-μm step size for eight steps were acquired at each time point. The photoactivation of optogenetic reporters was performed with an Andor Mosaic3 photo-stimulation module (Oxford Instruments) integrated into the SDCM system. For photoactivation, the time series was paused, an ROI was drawn on the interested part of the neutrophil, and a 445-nm (1.3 W) laser with 50% power and 3-s exposure was used to irradiate the ROI.

## Cytoplasmic calcium measurements

Cytosolic Ca$^{2+}$ was measured using the Fluo-4 Ca$^{2+}$ Imaging Kit (F10489; Invitrogen). 1 ml of dHL-60 cells at $5 × 10^5$ cells/ml were treated with 50 μM VU590 or DMSO control at 37°C for 2.5 h. The cells were then incubated with PowerLoad solution and Fluo-4 dye at 37°C for 15 min, followed by an additional 15 min at RT. After incubation, the cells were washed twice with mHBSS and then resuspended in 1 ml of mHBSS at a concentration of 5 × 10^5 cells/ml. The resuspension was then treated again with DMSO or the VU590 inhibitor. 200 μl of cells were loaded into fibrinogen-coated μ-Slide 8 Well High (80806; IBIDI) and left to settle at 37°C for 30 min before imaging. Green fluorescence images were recorded using a BioTek Lionheart FX Automated Microscope with a 20× phase lens at a 1-s interval for 10 seconds to obtain the baseline fluorescence. 20 μl of 1 μM fMLP was then added to the wells using a dispenser, and images were recorded for an additional 2 min with a 1-s interval. The resulting mean fluorescence intensity was normalized to that of the initial 10 s before fMLP addition.

## Image data analysis

Image analysis was performed using Imaris software and Fiji/ImageJ. For zebrafish neutrophil fluorescent image data, all images at the same time point of a single cell were projected to 2D for analysis. The cell images were segmented, and the background was subtracted in Imaris. The processed images were then imported into Fiji/ImageJ. Temporal color-coded cell outlines were plotted as described (Pal et al., 2023). Three spots were manually selected at both the leading and trailing edges of each neutrophil across all time points. The tracks for each spot over time were calculated using the built-in spots function in Imaris software. Protrusion and retraction speeds were then exported through the same function. The rolling Pearson's correlation was calculated with a five time frame window as previously described (Tsai et al., 2019). Line scans were performed in Fiji/ImageJ by drawing a straight-line segment within the cells, using a line of 12 pixels, to obtain an average intensity value. The values of biosensor fluorescent intensity were divided by that of mCherry or mCherry-CAAX to perform ratiometric analysis. The ratio of individual channel intensity profiles was smoothed using the Savitzky–Golay method and plotted in R. The kymograph was also plotted in R/RStudio after gathering all time points' biosensor/loading control normalized (min-max method) ratioed intensity.

## Statistical analysis

Statistical analysis was performed using Prism 6 (GraphPad). Normality is determined with the Shapiro–Wilk normality test and a QQ plot. Two-tailed Student's $t$ test and Mann–Whitney test were used to determine the statistical significance of differences between two groups when the data are normally or not normally distributed. One-way ANOVA and Kruskal–Wallis test were used to determine the statistical significance of multiple groups with normally or not normally distributed data. P value of $<0.05$ was considered statistically significant.

## Online supplemental material

Supplemental material provides additional figures and videos supporting the results. Specifically, Fig. S1 includes additional data that validate the selective role of *kcnj13* in zebrafish and mouse neutrophil migration. Fig. S2 validates the ASAP3 sensor in zebrafish and provides further data on the localization of other markers in zebrafish neutrophil chemotaxis. Fig. S3 illustrates the significance of the Gαi signal in depolarization-induced cell polarization and the cell's response to hyperpolarization. Video 1 shows PMN chemotaxis, complementing the cell tracks in Fig. 1. Video 2 shows neutrophil chemotaxis in the *jaguar* mutant, complementing the images in Fig. 2. Video 3 shows the polarity of MP in zebrafish neutrophil chemotaxis, complementing the images in Fig. 3. Video 4 illustrates the polarity of MP in zebrafish neutrophil random migration, complementing the images in Fig. S2. Video 5 shows the polarity of MP in zebrafish neutrophil chemotaxis upon Kir7.1 overexpression, complementing the images in Fig. 3. Video 6 demonstrates that photo-stimulation effectively regulates zebrafish neutrophil MP, complementing the images in Fig. 4. Videos 7, 8, and 9 show the cell response after photo stimulation at various cell location, complementing the images in Fig. 4. Video 10 shows the cell response after global hyperpolarization, complementing the images in Fig. S3.

## Acknowledgments

We thank Dr. Xiaoguang Zhu for his guidance on microscopes at the Bindley Bioscience Center, a core facility of the National Institutes of Health (NIH)-funded Indiana Clinical and Translational Sciences Institute. The HL-60 clonal cell line is a gift from Dr. Orion Weiner (University of California, San Francisco, CA, USA). The *jaguar*-like fish was a gift from Dr. David M. Parichy (University of Virginia, Charlottesville, VA, USA).

The work was supported by research funding from the NIH (R35GM119787 to Qing Deng, R35GM-124913 to GuangJun Zhang) and P30CA023168 to Purdue Center for Cancer Research for shared resources. This work is based upon efforts supported by EMBRIO Institute, contract #2120200, a National Science Foundation Biology Integration Institute.

Author contributions: Tianqi Wang: conceptualization, data curation, formal analysis, investigation, methodology, project administration, resources, software, validation, visualization, writing—original draft, review, and editing. Daniel H. Kim: data curation, formal analysis, investigation, validation, visualization, writing—review and editing. Chang Ding: formal analysis and investigation. Dingxun Wang: data curation and formal analysis. Weiwei Zhang: investigation and writing—original draft. Martin Silic: formal analysis, investigation, validation, and writing—review and editing. Xi Cheng: investigation. Kunming Shao: investigation. TingHsuan Ku: data curation, investigation, and methodology. Conwy Zheng: investigation and validation. Junkai Xie: investigation. Shulan Xiao: investigation. Krishna Jayant: supervision. Chongli Yuan: methodology. Alexander A. Chubykin: conceptualization, project administration, resources, supervision, and writing—review and editing. Christopher J. Staiger: conceptualization, funding acquisition, and project administration. GuangJun Zhang: conceptualization, funding acquisition, supervision, writing—original draft, review, and editing. Qing Deng: conceptualization, funding acquisition, project administration, resources, supervision, writing—original draft, review, and editing.

Disclosures: The authors declare no competing interests exist.

Submitted: 6 March 2025

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

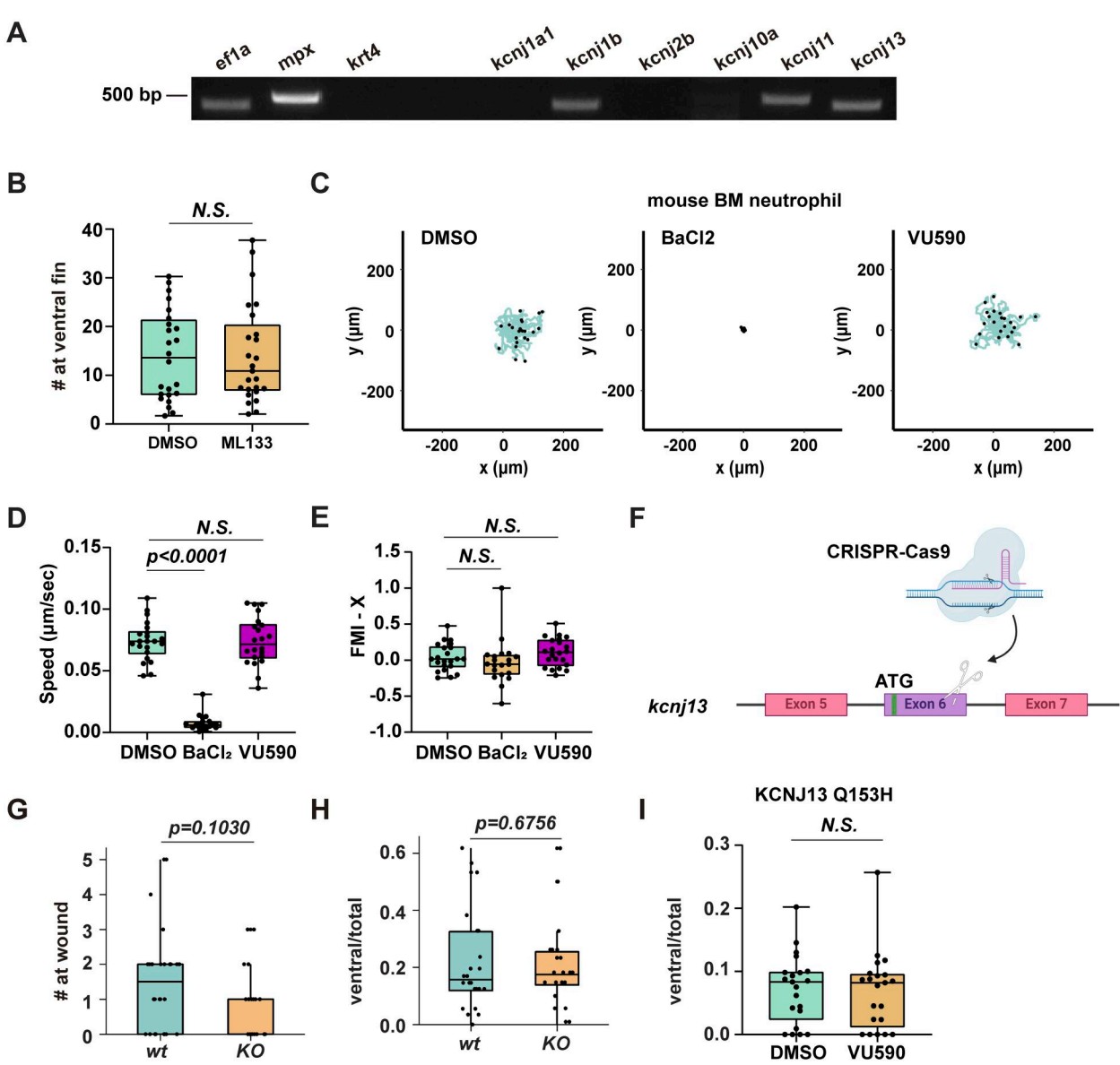

**Figure S1.** **Kir7.1 regulates zebrafish neutrophil chemotaxis. (A)** RT-PCR of indicated genes using mRNA extracted from FACS-sorted neutrophils. **(B)** Number of neutrophils recruited to the ventral fin induced by LTB4 in WT fish treated by ML133. **(C–E)** Representative tracks and quantification of speed and forward migration index (FMI) of mouse bone marrow PMN chemotaxis toward LTB4 after treatment with DMSO, BaCl$_2$, or VU590 in an under-agarose migration chamber plate coated with collagen IV. The lines represent individual cell trajectories over a 60-min period. $N > 15$ in each group. Data present mean ± SD, representative of three independent experiments. One-way ANOVA. **(F)** Schematic of the *kncj13* knockout fish. **(G and H)** Quantifications of neutrophils recruited to the tail wound (G) or ventral fin induced by LTB4 (H) in kcnj13 KO fish. **(I)** Quantification of neutrophils recruited to the ventral fin induced by LTB4 in neutrophil-specific KCNJ13$^{Q153H}$ mutant fish treated by VU590 or vehicle control. **(H and I)** The number of neutrophils in the fin was normalized to total neutrophil numbers to rule out developmental defects. **(B, G, H, and I)** Each dot represents one neutrophil. $n > 20$ in each group. Data representative of three independent experiments. Results are presented as mean ± SD, Mann–Whitney test. Source data are available for this figure: SourceData FS1.

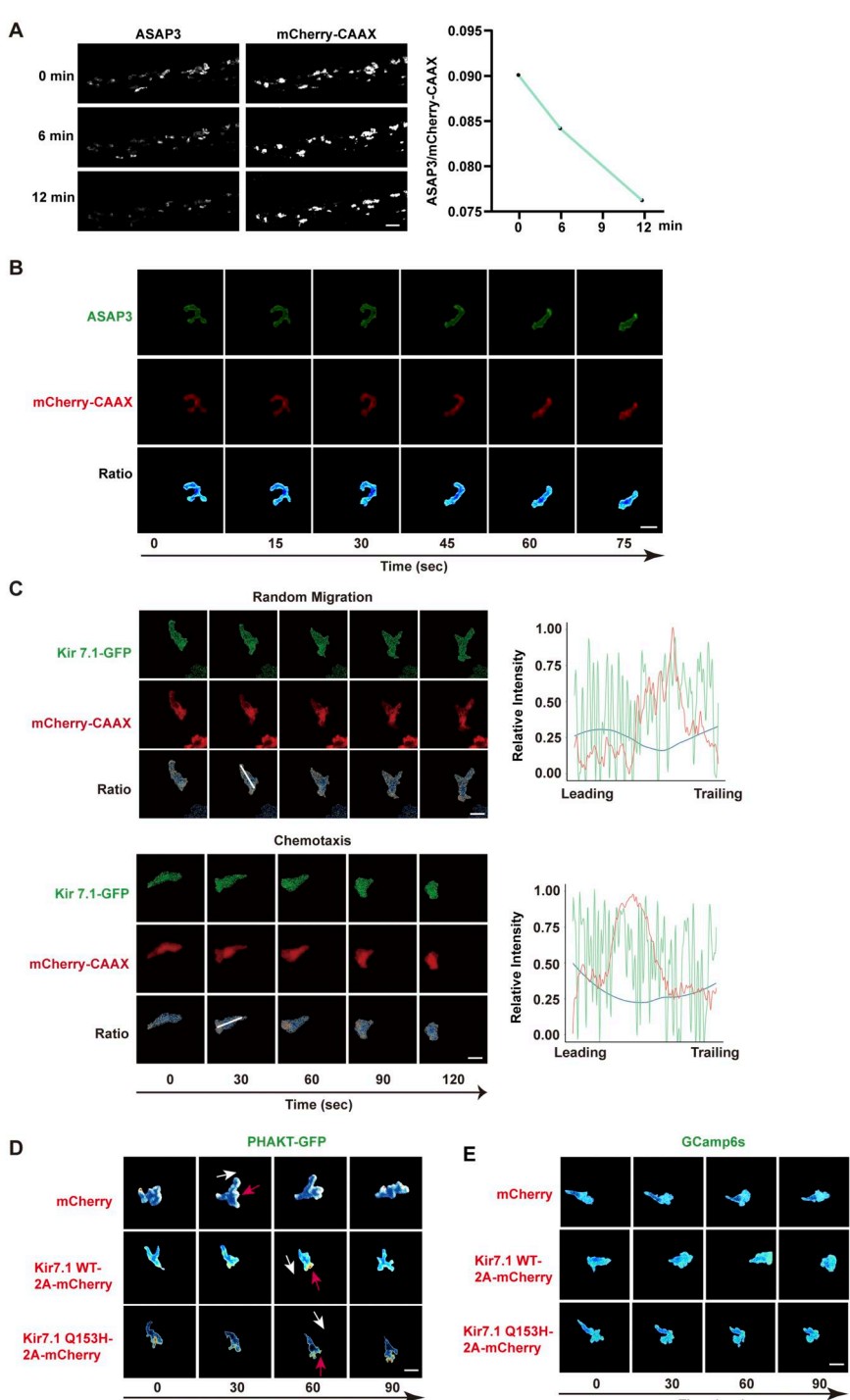

Figure S2. **Kir7.1 does not regulate cell polarity. (A)** Representative time-lapse imaging of ASAP3 and mCherry-CAAX in fish neutrophils at CHT treated with VU590 using *Tg(lyzC: ASAP3;lyzC: mCherry-CAAX)*. Ratiometric analysis (ASAP3/mCherry-CAAX) was performed to indicate a reduction in ASAP3 signal. Scale bars: 50 μm. Data representative of three independent experiments. **(B)** Representative time-lapse imaging of ASAP3 and mCherry-CAAX in fish neutrophils randomly migrating in the head mesenchyme using *Tg(lyzC: ASAP3;lyzC: mCherry-CAAX)*. Ratiometric analysis (ASAP3/mCherry-CAAX) was performed. Scale bars: 10 μm. Image representative of 15 neutrophils. **(C)** Representative time-lapse images of Kir 7.1-GFP and mCherry-CAAX during spontaneous migration in the head mesenchyme or chemotaxis to LTB4. Relative fluorescent intensity along the indicated line was plotted. Blue line indicates the ratio of GFP/mCherry. Images representative of 15 neutrophils for each condition. **(D)** Representative ratiometric images of PHAKT-GFP and mCherry during neutrophil migration to LTB4 in zebrafish *Tg(lyzC:mCherry;lyzC:PHART-GFP)*, *Tg(lyzC:kcnj13-2A-mCherry;lyzC:PHART-GFP)*, or *Tg(lyzC:kcnj13-Q153H-2A-mCherry;lyzC:PHAKT-GFP)*. The ratio indicates PHAKT-GFP/mCherry. White arrows indicate the direction of cell migration. Red arrows point to protrusions with a high accumulation of PHAKT. Scale bars: 10 μm. Image representatives of 15 neutrophils from each group. **(E)** Representative ratiometric images of GCamp6s and mCherry during neutrophils migration to LTB4 in zebrafish *Tg(lyzC:mCherry;lyzC:GCamp6s)*, *Tg(lyzC:kcnj13-2A-mCherry;lyzC: GCamp6s)*, or *Tg(lyzC:kcnj13-Q153H-2A-mCherry;lyzC: GCamp6s)*. The ratio indicates GCamp6S/mCherry. White arrows indicate the direction of cell migration. Red arrows point to areas with high-calcium flux. Scale bars: 10 μm. Image representatives of 15 neutrophils from each group.

JCB

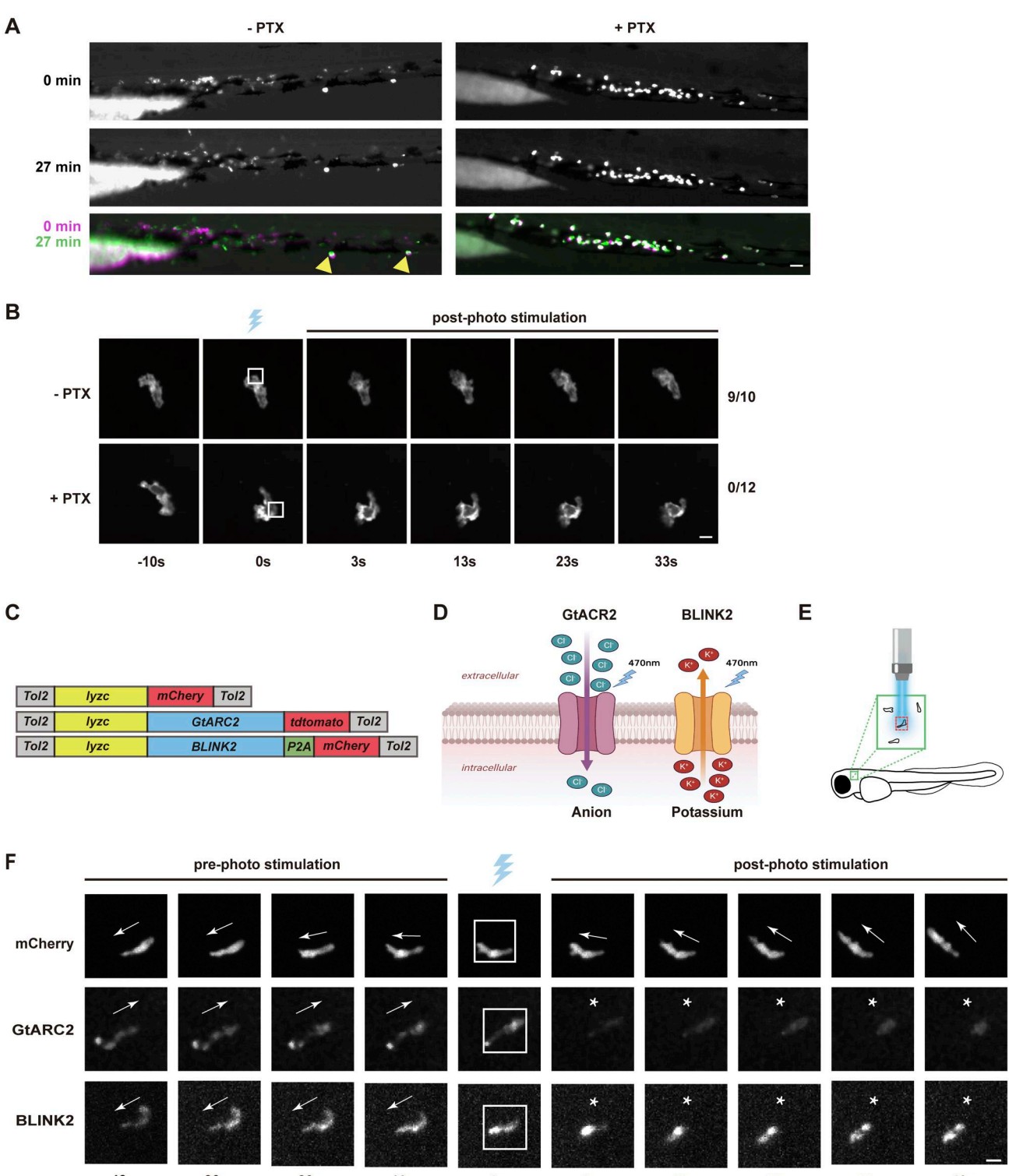

Figure S3. **Global plasma membrane depolarization stalls neutrophil migration. (A)** Representative images of neutrophil recruitment induced by LTB4 with or without PTX overexpression. Tol2-lyzC-GFP (–PTX) or Tol2-lyzC-GFP-2A-PTX (+PTX) was injected into 1-cell stage zebrafish embryos for transient expression. Neutrophil movement was indicated by overlaying the initial position with that at 27 min after LTB4 stimulation. Arrowheads label pigments. Scale bar, 50 µm. **(B)** Representative time-lapse images of neutrophils expressing CoChR with or without PTX overexpression before and after the laser stimulation at the indicated subcellular areas. Tol2-lyzC-GFP (–PTX) or Tol2-lyzC-GFP-2A-PTX (+PTX) was injected into 1-cell stage zebrafish embryos from *Tg(lyzC:CoChR-mCherry)^pu42* for transient expression. Scale bar, 10 µm. N_–PTX = 10 (9 generated protrusions); N_+PTX = 12 (0 generated protrusions). **(C and D)** Schematic of the construct design and ion selectivity of the optogenetic actuators. **(E)** Illustration of the neutrophils being imaged in the head mesenchyme. **(F)** Representative time-lapse images of neutrophil movement under the control of different optogenetic actuators or the mCherry control before and after stimulation using a 445 nm laser. The blue laser cartoon indicates the photo-stimulation frame. The white box indicates the stimulation area. White asterisks indicate the time frames during which migration was halted. Scale bar, 10 µm. *N* = 20 for each optogenetic manipulation group.

Video 1.  **Pharmacological inhibition of Kir7.1 during PMN chemotaxis toward fMLP.** Time-lapse imaging of human neutrophil chemotaxis toward fMLP in an under-agarose chemotaxis plate treated with DMSO, 40 mM BaCl₂, 40 mM KCl, 50 μM VU590, or 50 μM ML133. The migration trajectory is displayed over the tracked cell. Note, the circular ones are trapped in the gel, and only the ones spread on the plates were tracked. Scale bar: 100 μm.

Video 2.  **A point mutation in KCNJ13 decreases zebrafish neutrophil chemotaxis toward LTB4.** Time-lapse imaging of zebrafish neutrophil chemotaxis toward LTB4 in the ventral fin of the *Jaguar* mutant and the WT sibling control. The image started immediately after the LTB4 addition. The migration trajectory is displayed over the tracked cell. Scale bar: 200 μm.

Video 3.  **The polarity of MP in zebrafish neutrophil chemotaxis.** Ratiometric imaging of ASAP3 and mCherry-CAAX during zebrafish neutrophil chemotaxis towards LTB4 in the ventral fin in *Tg(lyzC: ASAP3;lyzC: mCherry-CAAX)*. The heatmap indicates a high-to-low ratio of ASAP3/mCherry. Scale bar: 10 μm.

Video 4.  **The polarity of the MP in zebrafish neutrophil random migration.** Ratiometric imaging of ASAP3 and mCherry-CAAX during neutrophil spontaneous migration in the head mesenchyme in *Tg(lyzC: ASAP3;lyzC: mCherry-CAAX)*. The heatmap indicates a high-to-low ratio of ASAP3/mCherry. Scale bar: 10 μm.

Video 5.  **The polarity of MP in zebrafish neutrophil chemotaxis upon Kir7.1 overexpression.** Ratiometric imaging of ASAP3 and mCherry-CAAX during zebrafish neutrophil chemotaxis toward LTB4 in the ventral fin in *Tg(lyzC: ASAP3;lyzC:kcnj13-2A-mCherry-CAAX)*(left) or *Tg(lyzC: ASAP3;lyzC:kcnj13-Q153H-2A-mCherry-CAAX)* (right). The heatmap indicates a high-to-low ratio of ASAP3/mCherry. Scale bar: 10 μm.

Video 6.  **Photo-stimulation effectively regulates zebrafish neutrophil MP.** Time-lapse imaging of zebrafish neutrophil spontaneous migration in the CHT before and after light stimulation. Neutrophils overexpress ASAP3 and CoCHR-mCherry-CAAX in *Tg(lyzC: ASAP3;lyzC:CoChR-mCherry*. Box indicates the frame and region of stimulation. Scale bar: 10 μm.

Video 7.  **Photo-stimulation biases pseudopod selection and neutrophil migration.** Time-lapse imaging of zebrafish neutrophil spontaneous migration in the head mesenchyme before and after light stimulation in *Tg(lyzC: CoChR-mCherry)^pu42*. Box indicates the frame and the pseudopod for stimulation. Scale bar: 10 μm.

Video 8.  **Photo-stimulation induces de novo protrusion during neutrophil migration.** Time-lapse imaging of zebrafish neutrophil spontaneous migration in the head mesenchyme before and after light stimulation in *Tg(lyzC: CoChR-mCherry)^pu42*. Box indicates the frame and the pseudopod for stimulation. Scale bar: 10 μm.

Video 9.  **Photo-stimulation at the tail of zebrafish neutrophils cannot reverse directionality.** Time-lapse imaging of zebrafish neutrophil spontaneous migration in the head mesenchyme before and after light stimulation in *Tg(lyzC: CoChR-mCherry)^pu42*. Box indicates the frame and the pseudopod for stimulation. Scale bar: 10 μm.

Video 10.  **Photo-stimulation during neutrophil spontaneous migration when cells overexpress GtARC2, BLINK2, or the mCherry control.** Time-lapse imaging of zebrafish neutrophil spontaneous migration in the head mesenchyme before and after light stimulation. Neutrophils overexpress GtARC2-mCherry, BLINK2-P2A-mCherry, or the mCherry control. The box indicates the frame and stimulation region. White asterisks indicate the time frames during which the migration halt occurred. Scale bar: 10 μm.

