## [Peer Review File · The Journal of Cell Biology]

Inwardly rectifying potassium channels promote directional sensing during neutrophil chemotaxis

Tianqi Wang, Daniel Kim, Chang Ding, Dingxun Wang, Weiwei Zhang, Martin Silic, Xi Cheng, Kunming Shao, TingHsuan Ku, Conwy Zheng, Junkai Xie, Shulan Xiao, Krishna Jayant, Chongli Yuan, Alexander Chubykin, Christopher Staiger, Guangjun Zhang, and Qing Deng

Corresponding Author(s): Qing Deng, Purdue University West Lafayette and Guangjun Zhang, Purdue University West Lafayette

Review Timeline:

Submission Date:	2025-03-06
Editorial Decision:	2025-04-11
Revision Received:	2025-08-15
Editorial Decision:	2025-09-23
Revision Received:	2025-10-01

Monitoring Editor: Anna Huttenlocher

Scientific Editor: Gabriele Stephan

Transaction Report:

DOI: <https://doi.org/10.1083/jcb.202503037>

April 11, 2025

Re: JCB manuscript #202503037

Qing Deng
Purdue University West Lafayette

Dear Dr. Deng,

Thank you for submitting your manuscript entitled "Inwardly rectifying potassium channels regulate direction sensing during neutrophil chemotaxis". The manuscript was assessed by expert reviewers, whose comments are appended to this letter. We invite you to submit a revision if you can address the reviewers' key concerns, as outlined here.

You will see that all reviewers seem enthusiastic about the paper, thinking that it potentially presents interesting findings regarding the membrane potential contribution to neutrophil chemotaxis. However, they all state that additional clarifications and data are needed in order to support claims in the manuscript, and we think that the concerns of all reviewers should be addressed. Particularly, the experiments suggested by reviewers #1 and #3 regarding the role of Kir7.1 in relation to other Kir family members and calibration/quantification of the ASAP3 probe as well as the potential discussion points suggested by Rev#2 would improve the manuscript.

GENERAL GUIDELINES:

Text limits: Character count for a Report is < 20,000, not including spaces. Count includes title page, abstract, introduction, the joint Results & Discussion, and acknowledgments. Count does not include materials and methods, figure legends, references, tables, or supplemental legends.

Figures: Reports may have up to 5 main text figures. To avoid delays in production, figures must be prepared according to the policies outlined in our Instructions to Authors, under Data Presentation, <https://jcb.rupress.org/site/misc/ifora.xhtml>. All figures in accepted manuscripts will be screened prior to publication.

Supplemental information: There are strict limits on the allowable amount of supplemental data. Reports may have up to 3 supplemental figures. Up to 10 supplemental videos or flash animations are allowed. A summary of all supplemental material should appear at the end of the Materials and methods section.

Please note that JCB now requires authors to submit Source Data used to generate figures containing gels and Western blots with all revised manuscripts. This Source Data consists of fully uncropped and unprocessed images for each gel/blot displayed in the main and supplemental figures. For assays performed using capillary electrophoresis and/or immunoassay-based detection, authors should instead provide the electropherogram graph(s) for each experiment, plotting fluorescence/chemiluminescence intensity vs. molecular weight/size. Please be sure to provide one Source Data file for each figure gels, blots, and/or capillary electrophoresis assays along with your revised manuscript files. File names for Source Data figures should be alphanumeric without any spaces or special characters (i.e., SourceDataF#, where F# refers to the associated main figure number or SourceDataFS# for those associated with Supplementary figures). For traditional gels and blots, the lanes of the gels/blots should be labeled as they are in the associated figure, the place where cropping was applied should be marked (with a box), and molecular weight/size standards should be labeled wherever possible. For capillary electrophoresis assays, each trace in the graph should be color-coded and labeled to indicate which protein, gene, or sample is being measured (please try to avoid red/green combinations to accommodate our color-blind readers).

The typical timeframe for revisions is three to four months. If you anticipate any difficulties in meeting this aforementioned revision time limit, please contact us and we can work with you to find an appropriate time frame for resubmission. Please note

that papers are generally considered through only one revision cycle, so any revised manuscript will likely be either accepted or rejected.

Thank you for this interesting contribution to Journal of Cell Biology. You can contact us at the journal office with any questions at cellbio@rockefeller.edu.

Sincerely,

Anna Huttenlocher
Monitoring Editor
Journal of Cell Biology

Gabriele Stephan, PhD
Scientific Editor
Journal of Cell Biology

Reviewer #1:

Wang et al. provide a potentially interesting manuscript, entitled "Inwardly rectifying potassium channels regulate direction sensing during neutrophil chemotaxis". The study touches on the controversial topic whether the MP can form a gradient along the plasma membrane and maintain this MP gradient in non-myelinated, "small" neutrophils during their chemotactic migration. This view on MP gradients in non-myelinated cells has been seen with skepticism in the electrophysiology field. Non-myelinated cells, e.g. a stimulated muscle cell, would transmit an action potential by continuous excitation conduction at a speed of ~1-20 meter per sec. This would mean that a cell of 10 micrometer size, such as a neutrophil, would be completely excited in < 1 ms or even much less. So there is the general question, whether a cell could maintain an MP gradient at the plasma membrane over time at all.

Based on genetic studies in zebrafish neutrophils and pharmacological inhibitor studies (using the compound VU590) in dHL-60 neutrophil-like cells and human neutrophils, the authors demonstrate a functional role of Kir7.1, an inwardly rectifying potassium channel, for maintaining the resting membrane potential of neutrophils (Fig. 5E). This has consequences for human neutrophils that are exposed to fMLP stimulation, leading to reduced downstream signaling (Fig. 5H, I) and impaired chemotactic responses (Fig. 5J). Surprisingly, the migration phenotype is restricted to chemotactic migration, observed in dHL-60 cells, human neutrophils and overexpressed Kir7.1 mutants in the fish (Fig. 1, 2 and 5A-C), but inhibition of Kir7.1 function does not interfere with random migration in the presence of stimulating GPCR ligand (Fig. 5D) or in the zebrafish tissue (Fig. 1C). By using the power of zebrafish genetics, the authors use the genetically encoded voltage indicator ASAP3 and suggest that the membrane potential (MP) is asymmetrically distributed between front and back of a chemotactic neutrophil, generating an MP gradient within the cell. To my knowledge, such spatiotemporal difference in MP has not been reported before in neutrophils. Hence, this could be potentially novel and interesting. Surprisingly, this gradient is not detectable in randomly migrating cells. Overexpression of wild-type Kir7.1 or the Kir7.1Q153H mutant in neutrophils leads to impaired chemotactic migration. Along that line, studies in Kir7.1G157E fish show comparable defects in chemotactic migration assays. Based on these studies, the authors argue that Kir7.1 regulates an MP gradient in neutrophils, which is likely critical for gradient sensing during chemotaxis. To demonstrate that the MP can directly affect cytoskeletal responses, the authors use optogenetic actuators in the fish system. Transgenic lines expressing CoCHR were used to locally depolarize neutrophils at the cell front, which strikingly results in the initiation of new neutrophil protrusions. A subcellular function of MP in migrating leukocytes has previously not been reported.

Overall, the study uses an impressive number of newly generated zebrafish strains to address the role of MP in the subcellular control of neutrophil chemotaxis. This aspect of neutrophil biology has conceptually not been fully worked out. The authors provide the interesting finding that Kir7.1 can regulate neutrophil chemotaxis. They also provide suggestive data that (1) an MP gradient might exist in chemotaxing neutrophils, and (2) the local activation of MP might trigger cell protrusion at the cell front. However, not all aspects of the study are entirely connected, a number of questions remained unanswered and some important technical control experiments were not included.

Major comments:

- 1) The authors show that treatment with the Kir inhibitor VU590 depolarizes dHL-60 cells, concluding that Kir channels maintain the resting membrane potential in dHL-60 cells (Fig. 5E). The use of VU590 leads to strong depolarization without the stimulus, and additional fMLP really does not change much. The authors measure a resting MP of -30mV, which is pretty "positive" and differs from MP measurements of human MP that were found to be at around -70mV. Oddly, the 40mM KCl control experiments lead to a depolarization to around 0mV, which does not really fit the potassium equilibrium potential at around -90mV (Messerer/Huber-Lang, 2018). This casts some doubts about the patch clamp measurements, but maybe the authors can explain these issues.
- 2) Unfortunately, there is a disconnect between the experiments in the fish and the data from dHL-60 cells. The fish experiments provide findings with the genetic ASAP3 probe, concluding that there might be an MP gradient in migrating neutrophils. Commonly, such genetic MP probes require calibration to assess their functionality, which is commonly done in cell experiments outside the living animal (e.g. Ruhl/Heinemann, Adv Sci 2024). While I understand that calibration experiments are difficult in the fish, it would be very critical to have a dataset in this study where the ASAP3 probe has been calibrated to draw solid conclusions.
 - A) The authors assess ASAP3 signal in relation to fluorescence membrane signal, which is a good attempt, but not fully conclusive. First, the calculation of the ASAP3/mCherry-CAAX ratio appears to be based on the analysis of a region through the cell interior (yellow line, Fig. 3H, I), but should rather be along the plasma membrane, right? Second, the asymmetric distribution of ASAP3 in cell front and back could result from processes involving an unequal distribution of ASAP3 in the cell. Hence, calibration experiments would be critical (see B).
 - B) Based on patch clamp techniques in dHL60 cells, the use of VU590 leads to strong depolarization without the stimulus, and fMLP really does not change much. This means that the membrane upon inhibition is already depolarized, right? Could the authors use dHL60 cells to express the ASAP3 probe in these cells and then perform (i) calibration experiments, and (ii) assess whether dHL60 cells also show an MP gradient under chemotactic, but not random migration conditions?
- 3) The authors show that VU590 depolarizes dHL-60 cells in the presence of homogeneous fMLP stimulation (independent of a chemotactic gradient), concluding that Kir channels maintain the resting membrane potential in dHL-60 cells. VU590 also impairs GPCR downstream signaling in response to fMLP. These findings occur in the presence of a homogenous GPCR stimulus. Hence, it is very surprising that random migration of zebrafish neutrophils in the tissue or dHL-60 cells in the presence of homogeneous fMLP do not show any impairment in their migration responses at all. How do the authors explain that? Obviously, randomly migrating cells do not require the maintenance of their MP, polarized membrane depolarization at leading edges or any form of MP gradient within the cells for efficient movement. But why? What is then mechanistically so special about the movement in the chemotactic gradient that Kir-dependent maintenance of MP would be required only under this condition? GPCR input signaling should not be substantially different between homogeneous or graded application of the GPCR stimulus.
- 4) The mechanistic link between GPCR signaling and maintenance of MP remained unclear to me. How are these two processes functionally interconnected? Is it still possible to photooptically activate protrusion formation by triggering membrane depolarization, when cells do not respond to GPCR signals (e.g., Galphai inhibition, such as pertussis toxin). Additionally, does photoactivation of membrane depolarization also trigger protrusion formation in dHL-60 cells?
- 5) Does the observed MP gradient in chemotactic neutrophil migration related to asymmetric distribution of the Kir7.1 channel under chemotactic, but not random migration conditions? Would immunostaining of the channel localization be feasible to address this point for unstimulated and stimulated condition?
- 6) It would be predictable that the Ca²⁺ influx is diminished upon Kir inhibition, but I am surprised that the Ca²⁺ gradient appears unaltered. If there is a localized Kir functionality and local change in MP, then this should also affect local Ca²⁺, right? Here is at least a discussion needed.
- 7) Experiments using the photooptical CoCH3 probe would also require some calibration experiments for MP changes to make sure that indeed local depolarization precedes protrusion formation. Otherwise, it could not be ruled out that other biological pathways, e.g. related to osmotic regulation, might be stimulated in this experimental setup and induce protrusion formation.

Minor comments:

- 1) Two mutants of Kir7.1 have been used in this study: Kir7.1G157E and Kir7.1Q153H. What are these mutants causing in the protein?
- 2) Could the authors provide more information on the ASAP3 sensor in text and figure. I assume that it senses hyperpolarization,

but this is not entirely obvious from the presented text and figure.

3) Movie 2 should show all individual channels, not only the the ratio of ASAP3 and mem-CAAX.

4) In line 91 they write that Masia et al 2015 did not pinpoint the actual rectifier, but actually they did according to the title and text, right?

Reviewer #2:

Wang et al. have provided a set of new data supporting membrane potential in chemotaxis, identifying inwardly rectifying potassium channels Kir 7.1. The data are well organized and clearly described. The manuscript is suitable for JCB audience and will add a significant understanding of bridging bioelectrical events in cells with better-studied chemotaxis. There are some suggestions I would like to make though.

1. Can the MP gradients be estimated through changes in mV or gradients of mV?

2. Although the global membrane potential (whole cell MP) may not be related directly to membrane charge (more local), when local MP is considered, the local changes in charge become important. The local change is what the authors focus on in the manuscript, so it is not correct, or at least not accurate to say " that MP regulation is separable from membrane charge".

Banerjee et al., 2022 paper suggests that "We speculate that some of these charge-sensitive components in turn initiate downstream events that mediate further loss of multiple anionic lipids in the front regions, further decreasing the membrane surface charge. Such feedback loops would enable small fluctuations to expand into propagating waves and can have outsized phenotypic effects. This architecture would be analogous to the ability of transmembrane potential to regulate key ion channels, which in turn regulate the transmembrane potential during action potential propagation."

3. Strong data suggest that membrane potential or its changes are not needed in the chemotaxis of Dictyostelium cells. I consider those experiments to be quite compelling, because cells after electroporation, in which large holes on the membrane were made, surprisingly, dicty cells still chemotaxis. Dicty cells are the most well-studied model of chemotaxis and share many key signaling pathways, including the more recently proposed "excitable networks" (Devreotes' work). How to reconcile these different requirement of membrane potentials in these two different types of cells will need to be discussed.

• J Cell Sci. 1990 Jan;95 (Pt 1):177-83. doi: 10.1242/jcs.95.1.177.PMID: 2161858

• Eukaryot Cell. 2011 Sep;10(9):1251-6.PMID: 21743003. PMCID: PMC3187056

4. Kir channels in migration have been studied in another type of directional migration, e.g. galvanotaxis. If the authors would like to bridge bioelectricity event with chemotaxis, these published results will be good to at least include in the discussion, i.e. Kir channels and their activities on membrane potentials in directional migration - those regulated by "bioelectricity" and by chemotaxis.

5. Directional sensing -regulation -

The title could be clearer through a more definitive description of "what type of regulation the Kir plays in chemotaxis, how-positive, negative.. facilitate?"

6. I find it puzzling that Kir7.1 regulates chemotaxis but not polarization. Polarization is the first step following directional sensing, then directional cell migration ensues. How exactly does Kir7.1 contribute to/ regulate chemotaxis?

7. VU590 selectively inhibits Kir1.1 (kcnj1) much more potent than kir 7.1, and primary human neutrophils express both coding genes KCNJ1 and kcnj13. The effects of UV590, therefore can not be exclusively attributed to Kir7.1.

8. "It is worth noting that MP regulation is separable from membrane charge (Banerjee et al., 2022) and galvanotaxis (migration in an electrical field)" - this statement does not have strong experimental evidence. The paper of Banerjee et al., 2022 does not exclude membrane potential in galvanotaxis. On the contrary, the references cited in Comment 1 above actually suggest MP is involved in galvanotaxis.

9. Polarized hyperpolarization --- reword

10. I do not see Fig. 7. Please check ---- "We randomly selected one protrusion to stimulate, and 8 out of 10 became dominant and guided cell migration (Fig. 7C and Movie 6)."

11. Also, check the following--

"The protrusion is relatively depolarized, and a slight hyperpolarization also correlates with the reduction in protrusion speed (Fig. 3G).

"The evident MP depolarization at the leading edge in control neutrophils was lost upon Kir7.1 wt or Q153H overexpression (Fig. 3G).

"We expressed the actuators in zebrafish neutrophils using transient plasmid injection and exposed the entire cell to blue light (Fig. 2C, D and Movies 9).

Minor:

1. Fig. 1C. Velocity is a vector quantity, meaning it requires both magnitude (speed) and direction to be fully defined, unlike speed which is a scalar. If the direction of cell migration is not analyzed here, "Speed" is a better word. Fig. 2F too.

2. Multiple mislabelled figures.

Reviewer #3:

Wang, Kim, Ding et al. investigate the role of the inwardly rectifying potassium channel Kir7.1 in regulating directional sensing during neutrophil chemotaxis. Using a combination of pharmacological, genetic, optogenetic, and imaging approaches in zebrafish and human neutrophils, the authors propose that Kir7.1 modulates spatial membrane potential gradients critical for gradient sensing, without affecting the basal motility or polarization machinery. They suggest that changes in membrane potential represent a novel regulatory layer in chemotaxis. While these findings are novel and offer a meaningful conceptual advance, further evidence clarifying the specific contribution of Kir7.1 versus other Kir family members would be necessary to strengthen the conclusions.

Major comments:

1. The authors convincingly show that Kir7.1 is expressed in neutrophils and that pharmacological inhibition via VU590 impairs directional chemotaxis toward LTB4 and wound cues. However, the specificity of VU590 (which targets both Kir1.1 and Kir7.1) remains a concern. Although genetic models are included, the Kir7.1 knockout fails to recapitulate the effects seen with VU590 and the dominant-negative mutants, suggesting possible functional redundancy. To support the central role of Kir7.1, it would be important to demonstrate that VU590 indeed alters resting membrane potential in zebrafish neutrophils (e.g. using measurements from the voltage indicator ASAP3 as a readout). Additionally, if Kir7.1 is the dominant regulator of resting MP, then VU590 should have a reduced or negligible effect in the *lyzC:kcnj13-Q153H* dominant-negative line. This experiment could clarify the channel's functional specificity.

2. Prior work by Masia et al., 2015 (*Am J Physiol Cell Physiol*) identified Kir2.1 as a major Kir channel in murine neutrophils, although its functional role was not explored. Cross-species evaluation of Kir2.1's contribution (e.g., using the relatively Kir2-selective inhibitor ML133 in zebrafish or human models) could strengthen the manuscript's mechanistic scope. Additionally, testing whether VU590 elicits similar chemotactic impairments in murine neutrophils would help establish the generalizability of the findings of the paper across species.

3. The measurement and quantification of membrane potential distributions using ASAP3 (Figure 3) require a better explanation: it is unclear how the ratiometric values in Fig. 3I, K, O, and S were calculated (especially when looking at Fig3I). Furthermore, Video 2, which presents an ASAP3/mCherry-CAAX ratio image, would benefit from including the raw green (ASAP3) and red (mCherry) channels, as is shown in Video 3. It would also be helpful to include a quantification of ASAP3 signal intensity in Fig. 4B to support the claim of localized depolarization following optogenetic stimulation.

Minor comments:

1. Video 9, related to Figure S3, should be revised - currently, the mCherry control neutrophil appears to stall after illumination, which contradicts the description in the text. Ideally, cells from all three experimental conditions should be shown side-by-side rather than sequentially, with clear coverage of both pre- and post-illumination periods in the same timelapse.

2. Several figures are misnumbered throughout the manuscript: for example, line 136 should refer to Fig. 1B and C; lines 212, 215, and 216 cite Fig. 7C, which does not exist, and line 223 should reference Fig. S3C and D.

3. In Figure 5B and 5C, the labels for DMSO and VU590 are missing.

4. Figure 3 panels H, I, J, and K are not referenced in the text, and the y-axis labeling in Fig. 3J appears incorrect and should be verified.

5. The manuscript would benefit from a round of careful proofreading (e.g line 55 contains the awkward phrase "Global hyperpolarizing neutrophils," and line 134 includes "where Kir regulates neutrophil migration," likely instead of „whether Kir regulates neutrophil migration", etc.)

We thank the reviewers for recognizing the novelty of our work and for the constructive comments and suggestions. We have conducted additional experiments and included further discussions. The manuscript is substantially improved as a result. Please see our point-by-point response below with the comments in bold.

Major comments:

1) The authors show that treatment with the Kir inhibitor VU590 depolarizes dHL-60 cells, concluding that Kir channels maintain the resting membrane potential in dHL-60 cells (Fig. 5E). The use of VU590 leads to strong depolarization without the stimulus, and additional fMLP really does not change much. The authors measure a resting MP of -30mV, which is pretty "positive" and differs from MP measurements of human MP that were found to be at around -70mV. Oddly, the 40mM KCl control experiments lead to a depolarization to around 0mV, which does not really fit the potassium equilibrium potential at around -90mV (Messerer/Huber-Lang, 2018). This casts some doubts about the patch clamp measurements, but maybe the authors can explain these issues.

HL-60 is a human tumor cell line that differentiates into neutrophil-like cells. The increased membrane potential may stem from the cancerous nature of the cells, as cancer cells tend to have a more depolarized V_m . The cell recordings vary widely, potentially reflecting differences in differentiation status. Nonetheless, recordings following VU590 or fMLP treatment are consistent, indicating clear depolarization. KCl depolarizes many cell types, including neurons and smooth muscle cells (Kirschstein et al., 2009; Rienecker et al., 2020). Elevated extracellular K^+ shifts the K^+ reversal potential, causing a net K^+ influx and a more positive (depolarized) membrane potential. The degree of depolarization, however, depends on the cell type. In addition to KCl, we used Ba^{2+} , a specific K^+ channel blocker that also induces depolarization. We observed a stronger phenotype with Ba^{2+} than with KCl, emphasizing the role of membrane potential in chemotaxis. These results are shown in Fig. 1J-K and Movie 1.

2) Unfortunately, there is a disconnect between the fish experiments and the data from dHL-60 cells. The fish studies utilize the genetic ASAP3 probe, suggesting a potential MP gradient in migrating neutrophils. Typically, such genetic MP probes require calibration to confirm their functionality, which is often done in cell experiments outside the living organism (e.g., Ruhl/Heinemann, Adv Sci 2024). While I understand that calibration in fish experiments is challenging, it would be critical to include a dataset within this study where the ASAP3 probe has been calibrated to support robust conclusions.

A) The authors assess ASAP3 signal in relation to fluorescence membrane signal, which is a good attempt, but not fully conclusive. First, the calculation of the ASAP3/mCherry-CAAX ratio appears to be based on the analysis of a region through the cell interior (yellow line, Fig. 3H, I), but should rather be along the plasma membrane, right?

Both ASAP3 and mCherry CAAX are localized to the plasma membrane, and the signals, even when they seem to be inside the cell, are membrane-bound. A similar approach, using a line scan across the cell, was used to quantify a membrane charge sensor (Banerjee et al., 2022).

Second, the asymmetric distribution of ASAP3 in cell front and back could result from processes involving an unequal distribution of ASAP3 in the cell. Hence, calibration experiments would be critical (see B).

ASAP family sensors are well-established and calibrated in cell cultures. ASAP3 increases brightness upon depolarization (Villette et al., 2019). ASAP3 is calibrated using $\% \Delta F/F_0$, (fluorescence intensity decreases as V_m rises (-70 to 0mV) compared to the initial intensity (F_0) at -70mV). Since neutrophils are moving, F_0 cannot be determined. We therefore normalized ASAP3 with mCherry-CAAX (plasma membrane volume control (Gong et al., 2024)). This technique is acceptable in vivo, as a similar approach is used to measure calcium dynamics in zebrafish neutrophils with GCAMP family sensors, which are also calibrated as $\Delta F/F_0$ (Beerman et al., 2015). As a result, we can only infer the gradient but not the exact value of V_m in our experiments.

Calibrating sensors in neutrophils within tissue is technically difficult. In vivo calibration of ASAP3 requires direct access to neutrophil membrane voltage, typically achieved through whole-cell patch clamping. However, in vivo patch clamping of neutrophils has never been reported before; therefore, we adapted neuron patch clamping techniques used in larval fishes. To approach neutrophils in the fin with patch pipettes, the fish skin has to be removed (Antinucci et al., 2020). This compromised the fin's tissue integrity, causing the tissue to become loose and resulting in some neutrophils losing their tight embedment. During patch clamping attempts, we successfully attached to these neutrophils and reached gigaohm seal configurations, but faced difficulties breaking open the membranes without damaging the cells or detaching the pipettes. This was likely due to the absence of a firm substrate caused by skin removal. Even when membrane breaking was successful, maintaining stable recordings in neutrophils was unsuccessful due to rapid cell detachment.

The rEstus (Ruhl/Heinemann, Adv. Sci. 2024) is an excitation ratiometric sensor (480/400 nm). Since it uses two different excitation wavelengths, this sensor is not suitable for imaging V_m dynamics in fast-moving cells because of inevitable delays when capturing the two channels. In our experiments, we acquired both the ASAP3/mCherry channels simultaneously.

B) Based on patch clamp techniques in dHL60 cells, the use of VU590 leads to strong depolarization without the stimulus, and fMLP really does not change much. This means that the membrane upon inhibition is already depolarized, right? Could the authors use dHL60 cells to express the ASAP3 probe in these cells and then perform (i) calibration experiments, and (ii) assess whether dHL60 cells also show an MP gradient under chemotactic, but not random migration conditions?

We agree with the reviewer that VU590 depolarizes the cell to a level similar to that of fMLP. We attempted to express ASAP3 in HL-60 cells; however, the probe is trapped in the endosomal compartment. We have included an example figure showing the localization of ASAP3 in dHL-60 cells (ASAP/Wheat germ agglutinin (WGA) for general membrane labeling/bright field). We do not understand the cause, but various trapping

in endosomal compartments is observed when ASAP3 is fused with red fluorescent proteins, which, in theory, should not affect membrane trafficking (Figure 3 of (Kim et al., 2022)).

3) The authors show that VU590 depolarizes dHL-60 cells in the presence of homogeneous fMLP stimulation (independent of a chemotactic gradient), concluding that Kir channels maintain the resting membrane potential in dHL-60 cells. VU590 also impairs GPCR downstream signaling in response to fMLP. These findings occur in the presence of a homogenous GPCR stimulus. Hence, it is very surprising that random migration of zebrafish neutrophils in the tissue or dHL-60 cells in the presence of homogeneous fMLP do not show any impairment in their migration responses at all. How do the authors explain that? Obviously, randomly migrating cells do not require the maintenance of their MP, polarized membrane depolarization at leading edges or any form of MP gradient within the cells for efficient movement. But why? What is then mechanistically so special about the movement in the chemotactic gradient that Kir-dependent maintenance of MP would be required only under this condition? GPCR input signaling should not be substantially different between homogeneous or graded application of the GPCR stimulus.

Cell migration relies on inherent cell excitability, driven by spontaneous self-organization of the phosphatidylinositol lipid signaling system and Ras signaling (Arai et al., 2010; Matsuoka et al., 2024). Chemokines temporally and spatially influence the excitability of the network to direct chemotaxis or directional migration (Xiong et al., 2010). Therefore, our data indicate that MP is crucial for chemokine sensing but not for intrinsic cell excitability.

A homogeneous GPCR stimulus induces a temporary increase in intracellular signaling, but it leads to ongoing chemokinesis. The exact process behind chemokinesis is not fully understood. We propose that VU590 simulates the uniform chemokine treatment, which triggers brief signaling activation, long-term suppression, and continuous chemokinesis. As a result, the cells become insensitive to further chemotaxis signals.

The mechanism of zebrafish neutrophil random migration in the head is unclear. It is unknown whether a chemokine is present in the head to drive chemokinesis. If so, the data on fish and dHL-60 are consistent: random migration in the presence of a chemokine (chemokinesis) is independent of MP and unaffected by VU590 treatment.

4) The mechanistic link between GPCR signaling and maintenance of MP remained unclear to me. How are these two processes functionally interconnected? Is it still possible to photooptically activate protrusion formation by triggering membrane depolarization, when cells do not respond to GPCR signals (e.g., Galphai inhibition, such as pertussis toxin). Additionally, does photoactivation of membrane depolarization also trigger protrusion formation in dHL-60 cells?

Since pertussis toxin cannot penetrate tissue, we expressed the catalytic domain of PTX in zebrafish neutrophils (Hammerschmidt and McMahon, 1998). Zebrafish neutrophils expressing PTX are unable to respond to LTB₄, which is consistent with impaired GPCR signaling. The CoCHR photoactivation also failed to induce protrusion in the presence of PTX, indicating that Gai/o is necessary for CoCHR-induced locomotion. The data are now included in Fig. S3A-B.

Unfortunately, we lack the equipment to perform photoactivation in dHL-60 cells. The microscope cannot support a stage-top incubation system; therefore, we cannot carry out photoactivation at 37°C, which is essential for mammalian cells. We plan to try this experiment in the future.

5) Does the observed MP gradient in chemotactic neutrophil migration related to assymmetric distribution of the Kir7.1 channel under chemotactic, but not random migration conditions? Would immunostaining of the channel localization be feasible to address this point for unstimulated and stimulated condition?

We were unable to find a suitable antibody for immunostaining in dHL-60 cells. Instead, we expressed Kir7.1-GFP fusion in zebrafish neutrophils. We also observe asymmetric channel localization in fish neutrophils during chemotaxis, but not during random migration. The data is now included in Fig. S2C.

6) It would be predictable that the Ca²⁺ influx is diminished upon Kir inhibition, but I am surprised that the Ca²⁺ gradient appears unaltered. If there is a localized Kir functionality and local change in MP, then this should also affect local Ca²⁺, right? Here is at least a discussion needed.

MP can regulate Ca²⁺ entry at the plasma membrane, especially through store-operated calcium entry (Kozak and Putney, 2017). Our results with high-level uniform fMLP stimulation in dHL-60 align with the physics.

In zebrafish neutrophil chemotaxis, calcium imaging reflects the gradient rather than the absolute amount of calcium. The calcium levels between different cells cannot be directly compared due to varying biosensor expression levels. While the overall cell calcium level may change, the gradient remains unaffected. Cytoplasmic calcium can also originate from the ER store, which is not regulated by the MP. We observed minimal calcium gradients throughout the experiments. This may be due to the rapid diffusion of calcium in the cytoplasm. The discussion is now included in the manuscript.

7) Experiments using the photooptical CoCH3 probe would also require some calibration experiments for MP changes to make sure that indeed local depolarization precedes protrusion formation. Otherwise, it could not be ruled out that other biological pathways, e.g. related to osmotic regulation, might be stimulated in this experimental setup and induce protrusion formation.

Determining local depolarization before protrusion formation is technically challenging. CoCH3 becomes inactive 20-50ms after the laser is turned off. Electrical signals fade too quickly for us to capture local depolarization using our current equipment, as discussed in the manuscript and illustrated in Fig. 4B. Another limitation of the ASAP family sensors is that they are not suitable for high-frequency imaging due to photobleaching. We will attempt to use Voltron2 (Abdelfattah et al., 2023), which is more photostable than ASAP3, in our follow-up studies, utilizing high-frequency imaging. Osmotic regulation is unlikely to cause protrusion formation, as opening K⁺ channels causes cells to contract rather than protrude (Fig. S3).

Minor comments:

1) Two mutants of Kir7.1 have been used in this study: Kir7.1G157E and Kir7.1Q153H. What are these mutants causing in the protein?

Both mutations cause a loss of function. Since Kir channels function as tetramers, their conductance decreases when composed of both wild-type and mutant proteins due to a stoichiometric effect. We used zebrafish expressing Kir7.1G157E, which display a Jaguar-like pigment pattern (Iwashita et al., 2006; Silic et al., 2020), and this model is available to us. Kir7.1Q153H has been confirmed to reduce K⁺ conductance in HEK293T cells when overexpressed (Silic et al., 2020).

2) Could the authors provide more information on the ASAP3 sensor in text and figure. I assume that it senses hyperpolarization, but this is not entirely obvious from the presented text and figure.

ASAP family sensors are well-established and calibrated in cell cultures. ASAP3 increases brightness upon depolarization (Villette et al., 2019). ASAP3 is calibrated using $\% \Delta F/F_0$, where fluorescence intensity decreases as V_m rises from -70 to 0 mV compared to the initial intensity at -70 mV. We have added this information to the main text.

3) Movie 2 should show all individual channels, not only the ratio of ASAP3 and mem-CAAX.

We have included the individual channels in the movies.

4) In line 91 they write that Masia et al 2015 did not pinpoint the actual rectifier, but actually they did according to the title and text, right?

Masia et al., 2015, showed that the rectifying current is blocked by Ba(2+), Cs(+), and the Kir2-selective inhibitor ML133. Although the results support Kir2.1 as the main rectifier, they are not confirmed with Kir2.1 knockout and are therefore suggestive but not definitive. We have updated the text to reflect the background information.

Reviewer #2:

1. Can the MP gradients be estimated through changes in mV or through gradients of mV?

We cannot estimate the MP gradient in vivo due to the difficulty in calibration. Please see our response to Reviewer 1, also copied below:

Measurements and calibrations must be performed in the same cell to obtain an accurate measurement because the varying expression levels of sensors and imaging acquisition conditions can influence the results. Consequently, after the initial calibration during sensor development, most in vivo experiments are conducted without further calibration. Calibrating the

sensor in neutrophils within tissue is technically difficult. For in vivo calibration, fish skin needs to be removed (Antinucci et al., 2020). We attempted to calibrate but were unsuccessful.

2. Although the global membrane potential (whole cell MP) may not be related directly to membrane charge (more local), when local MP is considered, the local changes in charge become important. The local change is what the authors focus on in the manuscript, so it is not correct, or at least not accurate to say " that MP regulation is separable from membrane charge". Banerjee et al., 2022 paper suggests that "We speculate that some of these charge-sensitive components in turn initiate downstream events that mediate further loss of multiple anionic lipids in the front regions, further decreasing the membrane surface charge. Such feedback loops would enable small fluctuations to expand into propagating waves and can have outsized phenotypic effects. This architecture would be analogous to the ability of transmembrane potential to regulate key ion channels, which in turn regulate the transmembrane potential during action potential propagation."

The discussion referenced in the Banerjee et al., 2022 paper speculated that inner membrane charge can propagate and self-amplify like an action potential. While inner membrane charge can contribute to MP, they are two different phenomena. MP mainly focuses on the charge differences across the thin plasma membrane. The inner membrane charge primarily attracts molecules of the opposite charge, whereas MP mostly regulates the conformation of membrane proteins.

3. Strong data suggest that membrane potential or its changes are not needed in the chemotaxis of Dictyostelium cells. I consider those experiments to be quite compelling, because cells after electroporation, in which large holes on the membrane were made, surprisingly, dicty cells still chemotaxis. Dicty cells are the most well-studied model of chemotaxis and share many key signaling pathways, including the more recently proposed "excitable networks" (Devreotes' work). How to reconcile these different requirement of membrane potentials in these two different types of cells will need to be discussed.
• J Cell Sci. 1990 Jan;95 (Pt 1):177-83. doi: 10.1242/jcs.95.1.177.PMID: 2161858

While Dicty cells are a well-established model for chemotaxis, the membrane potential did not influence Dicty motility or chemotaxis to cAMP (Gao et al., 2011; Van Duijn et al., 1990). The finding that MP is not necessary for cell movement is also observed in zebrafish neutrophils and human dHL-60 cells. However, the role of MP in directional sensing varies among these systems. This difference may result from variations in GPCR signaling in Dicty and neutrophils. In Dicty, cAMP activates $G\alpha_{2\beta}$, leading to cAMP/cGMP production, whereas in neutrophils, cAMP is not generated via GPCR- $G\alpha_i$ signaling. Additionally, V_m in Dicty is maintained by an electrogenic proton pump, unlike the K^+/Na^+ pumps in neutrophils. Most importantly, in systems where V_m is crucial for chemotaxis, such as neutrophils and *Physarum polycephalum* (Ueda et al., 1975)(<https://doi.org/10.1007/BF01290773>), chemotactic factors induce ion fluxes and changes in membrane potential. It remains unclear whether cAMP causes depolarization in Dicty cells. We speculate that GPCR-induced depolarization correlates with the functional importance of V_m in chemotaxis. The discussion has been revised to include this point.

4. Kir channels in migration have been studied in another type of directional migration, e.g. galvanotaxis. If the authors would like to bridge bioelectricity event with chemotaxis, these published results will be good to at least include in the discussion, i.e. Kir channels and their activities on membrane potentials in directional migration - those regulated by "bioelectricity" and by chemotaxis.

In addition to Dicty (as mentioned above), in epithelial cells, KCNJ15 (encoding Kir4.2) knockdown specifically abolishes directionality, not speed, during galvanotaxis, without affecting basal motility and directional migration in a monolayer scratch assay (Nakajima et al., 2015). Therefore, Kir channels may regulate membrane potential during both galvanotaxis and chemotaxis, although the specific channels and biological processes involved are cell-type dependent. We have included this point in the discussion.

5. Directional sensing regulation

The title could be clearer with a more definitive description of what type of regulation the Kir channels play in chemotaxis, such as whether it is positive or negative, and how they facilitate this process.

We reflect the positive regulation: Inwardly rectifying potassium channels promote directional sensing during neutrophil chemotaxis

6. I find it puzzling that Kir7.1 regulates chemotaxis but not polarization. Polarization is the first step following directional sensing, then directional cell migration ensues. How exactly does Kir7.1 contribute to/ regulate chemotaxis?

Cell migration relies on the intrinsic excitability of cells, driven by the spontaneous self-organization of the phosphatidylinositol lipid signaling system and Ras signaling (Arai et al., 2010; Matsuoka et al., 2024). Chemokines spatiotemporally bias the network's excitability to guide chemotaxis or directional migration (Xiong et al., 2010). Therefore, our data suggest that MP is essential for chemokine sensing but not for intrinsic cell excitability or polarization.

We hypothesize that VU590 bath mimics the uniform chemokine treatment, which induces transient signaling activation and persistent chemokinesis. The cells then become insensitive to additional chemotaxis cues, so VU590 only blocks chemotaxis, not random migration. We also obtained additional data (Fig. S3A, B) showing that depolarization-induced protrusion depends on Gai signaling, further supporting the connection between membrane potential and GPCR crosstalk.

7. VU590 selectively inhibits Kir1.1 (kcnj1) much more potent than kir 7.1, and primary human neutrophils express both coding genes KCNJ1 and kcnj13. The effects of UV590, therefore can not be exclusively attributed to Kir7.1.

We agreed with the reviewer. We were careful when discussing our results and writing the manuscript, using “Inwardly rectifying potassium channels” rather than “kir 7.1” in our title. This is primarily because of the expression of different Kir family members in zebrafish and dHL-60 cells.

8. "It is worth noting that MP regulation is separable from membrane charge (Banerjee et al., 2022) and galvanotaxis (migration in an electrical field)" - this statement does not have strong experimental evidence. The paper of Banerjee et al., 2022 does not exclude membrane potential in galvanotaxis. On the contrary, the references cited in Comment 1 above actually suggest MP is involved in galvanotaxis.

We apologize for the confusion. We did not mean that MP would be dispensable for galvanotaxis. Although related, we wanted to highlight that membrane charge and MP are connected to cell physics and influence cell processes through separate mechanisms. The electric field is expected to change the membrane potential, which is necessary for galvanotaxis. Besides this mechanism, electrical fields also attract charged molecules (Belliveau et al., 2024). We have updated the discussion.

9. Polarized hyperpolarization — reword

We mean the tails are more hyperpolarized. We rephrased to “hyperpolarization” in the abstract.

10. I do not see Fig. 7. Please check ---- "We randomly selected one protrusion to stimulate, and 8 out of 10 became dominant and guided cell migration (Fig. 7C and Movie 6)."

11. Also, check the following--

"The protrusion is relatively depolarized, and a slight hyperpolarization also correlates with the reduction in protrusion speed (Fig. 3G).

"The evident MP depolarization at the leading edge in control neutrophils was lost upon Kir7.1 wt or Q153H overexpression (Fig. 3G).

"We expressed the actuators in zebrafish neutrophils using transient plasmid injection and exposed the entire cell to blue light (Fig. 2C, D and Movies 9).

I apologize for the oversight. We have thoroughly reviewed the manuscript to confirm that the figures and movies are correctly cited.

Minor:

1. Fig. 1C. Velocity is a vector quantity, meaning it requires both magnitude (speed) and direction to be fully defined, unlike speed which is a scalar. If the direction of cell migration is not analyzed here, "Speed" is a better word. Fig. 2F too.

2. Multiple mislabelled figures.

We have updated the manuscript and used 'speed' instead of 'velocity'.

Reviewer #3:

Wang, Kim, Ding et al. investigate the role of the inwardly rectifying potassium channel Kir7.1 in regulating directional sensing during neutrophil chemotaxis. Using a combination of pharmacological, genetic, optogenetic, and imaging approaches in zebrafish and human neutrophils, the authors propose that Kir7.1 modulates spatial membrane potential

gradients critical for gradient sensing, without affecting the basal motility or polarization machinery. They suggest that changes in membrane potential represent a novel regulatory layer in chemotaxis. While these findings are novel and offer a meaningful conceptual advance, further evidence clarifying the specific contribution of Kir7.1 versus other Kir family members would be necessary to strengthen the conclusions.

Major comments:

1. The authors convincingly show that Kir7.1 is expressed in neutrophils and that pharmacological inhibition via VU590 impairs directional chemotaxis toward LTB4 and wound cues. However, the specificity of VU590 (which targets both Kir1.1 and Kir7.1) remains a concern. Although genetic models are included, the Kir7.1 knockout fails to recapitulate the effects seen with VU590 and the dominant-negative mutants, suggesting possible functional redundancy. To support the central role of Kir7.1, it would be important to demonstrate that VU590 indeed alters resting membrane potential in zebrafish neutrophils (e.g. using measurements from the voltage indicator ASAP3 as a readout). Additionally, if Kir7.1 is the dominant regulator of resting MP, then VU590 should have a reduced or negligible effect in the *lyzC:kcj13-Q153H* dominant-negative line. This experiment could clarify the channel's functional specificity.

We thank the reviewer for the suggestion. We visualized ASAP3 in zebrafish neutrophils after VU590 treatment and indeed saw a decrease in the ASAP3/mCherry-CAAX ratio, indicating that VU590 caused depolarization. This data is now included in Fig. S2A. We also treated the *lyzC:kcj13-Q153H* dominant-negative line with VU590. We did not observe additional inhibition of chemotaxis, suggesting that VU590 mainly targets Kir7.1. This data is now included in Fig. S1I. Please note that it is common for a knockout to not fully replicate the phenotype of a knockdown or dominant-negative, due to potential genetic compensation (upregulation of similar genes).

2. Prior work by Masia et al., 2015 (Am J Physiol Cell Physiol) identified Kir2.1 as a major Kir channel in murine neutrophils, although its functional role was not explored. Cross-species evaluation of Kir2.1's contribution (e.g., using the relatively Kir2-selective inhibitor ML133 in zebrafish or human models) could strengthen the manuscript's mechanistic scope. Additionally, testing whether VU590 elicits similar chemotactic impairments in murine neutrophils would help establish the generalizability of the findings of the paper across species.

We performed the experiments as suggested. ML-133 does not affect zebrafish neutrophil chemotaxis. The results are now included in Fig. S1B. This is consistent with the data showing that zebrafish neutrophils do not express Kir2.1. ML-133 affected human primary neutrophil chemotaxis to a similar degree as VU590. Human neutrophils express (from high to low): KCNJ15/2/1/13/8/14/12/11/16/3/6/10/5/9. The results are now shown in Fig. 1J-L. We also treated mouse bone marrow neutrophils with VU590 and did not observe an inhibition in chemotaxis. Mouse neutrophils express (from high to low): KCNJ8/10/5/13/16/14/9/15/1. These results are now included in Fig. S1C-E. The expanded and diverse expression pattern of KCNJ

family genes in mouse and human further highlights their physiological importance. Although a reduction in directional sensing is conserved across systems, the effect of V_m on speed is not conserved between species, possibly due to differences in the degree of V_m inhibition.

For reviewer's reference: VU590 is a small molecule inhibitor of KCNJ1 (IC₅₀ = 294 nM), which also causes 70% inhibition of KCNJ13 at 10 μ M. It does not affect the related channels KCNJ2 (Kir2.1) and KCNJ10 (Kir4.1) at a concentration of 10 μ M.

ML 133 hydrochloride is a selective blocker of inwardly rectifying Kir2 potassium channels (IC₅₀ values are 1.8, 2.8, 2.9, and 4.0 μ M at pH 7.4 for mKir2.1, hKir2.6, hKir2.2, and hKir2.3, respectively). Exhibits no effect on rKir1.1 (IC₅₀ > 300 μ M); displays weak activity at hKir7.1 and rKir4.1 (IC₅₀ values are 33 and 76 μ M).

3. The measurement and quantification of membrane potential distributions using ASAP3 (Fig. 3) require a better explanation: it is unclear how the ratiometric values in Fig. 3I, K, O, and S were calculated (especially when looking at Fig. 3I). Furthermore, Video 2, which presents an ASAP3/mCherry-CAAX ratio image, would benefit from including the raw green (ASAP3) and red (mCherry) channels, as is shown in Video 3. It would also be helpful to include a quantification of ASAP3 signal intensity in Fig. 4B to support the claim of localized depolarization following optogenetic stimulation.

We divided ASAP3/mCherry CAAX. We have included the raw divided data (the black curve) for Fig. 3I. The black curve is then smoothed to generate the blue curve, which is included in the figure.

We have separated the channels for Video 2.

We have added a quantification of the ASAP3/mCherry to Fig. 4B to show the transient depolarization. Please note that we were unable to observe localized depolarization, possibly because our imaging frequency was too slow to capture it before the spread of the electrical signals.

Minor comments:

1. Video 9, related to Figure S3, should be revised - currently, the mCherry control neutrophil appears to stall after illumination, which contradicts the description in the text. Ideally, cells from all three experimental conditions should be shown side-by-side rather than sequentially, with clear coverage of both pre- and post- illumination periods in the same time lapse.

We have updated Video 9 to display three experimental conditions side by side.

2. Several figures are misnumbered throughout the manuscript: for example, line 136 should refer to Fig. 1B and C; lines 212, 215, and 216 cite Fig. 7C, which does not exist, and line 223 should reference Fig. S3C and D.

I apologize for the oversight. We have carefully reviewed the manuscript to ensure that the figures and movies are correctly cited.

3. In Figure 5B and 5C, the labels for DMSO and VU590 are missing.

We have added the labels to the figure.

4. Figure 3 panels H, I, J, and K are not referenced in the text, and the y-axis labeling in Fig. 3J appears incorrect and should be verified.

I apologize for the oversight. We have thoroughly reviewed the manuscript to ensure that the figures and movies are properly cited.

5. The manuscript would benefit from a round of careful proofreading (e.g line 55 contains the awkward phrase "Global hyperpolarizing neutrophils," and line 134 includes "where Kir regulates neutrophil migration," likely instead of „whether Kir regulates neutrophil migration", etc.)

We have completed several rounds of proofreading.

Reference:

- Abdelfattah, A.S., J. Zheng, A. Singh, Y.C. Huang, D. Reep, G. Tsegaye, A. Tsang, B.J. Arthur, M. Rehorova, C.V.L. Olson, Y. Shuai, L. Zhang, T.M. Fu, D.E. Milkie, M.V. Moya, T.D. Weber, A.L. Lemire, C.A. Baker, N. Falco, Q. Zheng, J.B. Grimm, M.C. Yip, D. Walpita, M. Chase, L. Campagnola, G.J. Murphy, A.M. Wong, C.R. Forest, J. Mertz, M.N. Economo, G.C. Turner, M. Koyama, B.J. Lin, E. Betzig, O. Novak, L.D. Lavis, K. Svoboda, W. Korff, T.W. Chen, E.R. Schreiter, J.P. Hasseman, and I. Kolb. 2023. Sensitivity optimization of a rhodopsin-based fluorescent voltage indicator. *Neuron*. 111:1547-1563 e1549.
- Antinucci, P., A. Dumitrescu, C. Deleuze, H.J. Morley, K. Leung, T. Hagley, F. Kubo, H. Baier, I.H. Bianco, and C. Wyart. 2020. A calibrated optogenetic toolbox of stable zebrafish opsin lines. *eLife*. 9.
- Arai, Y., T. Shibata, S. Matsuoka, M.J. Sato, T. Yanagida, and M. Ueda. 2010. Self-organization of the phosphatidylinositol lipids signaling system for random cell migration. *Proceedings of the National Academy of Sciences of the United States of America*. 107:12399-12404.
- Banerjee, T., D. Biswas, D.S. Pal, Y. Miao, P.A. Iglesias, and P.N. Devreotes. 2022. Spatiotemporal dynamics of membrane surface charge regulates cell polarity and migration. *Nature cell biology*. 24:1499-1515.
- Beerman, R.W., M.A. Matty, G.G. Au, L.L. Looger, K.R. Choudhury, P.J. Keller, and D.M. Tobin. 2015. Direct In Vivo Manipulation and Imaging of Calcium Transients in Neutrophils Identify a Critical Role for Leading-Edge Calcium Flux. *Cell reports*. 13:2107-2117.
- Belliveau, N.M., M.J. Footer, A. Platenkamp, H. Kim, T.E. Eustis, and J.A. Theriot. 2024. Galvanin is an electric-field sensor for directed cell migration. *bioRxiv*.
- Gao, R.C., X.D. Zhang, Y.H. Sun, Y. Kamimura, A. Mogilner, P.N. Devreotes, and M. Zhao. 2011. Different roles of membrane potentials in electrotaxis and chemotaxis of dictyostelium cells. *Eukaryotic cell*. 10:1251-1256.
- Gong, B., J.D. Johnston, A. Thiemicke, A. de Marco, and T. Meyer. 2024. Endoplasmic reticulum-plasma membrane contact gradients direct cell migration. *Nature*. 631:415-423.

- Hammerschmidt, M., and A.P. McMahon. 1998. The effect of pertussis toxin on zebrafish development: a possible role for inhibitory G-proteins in hedgehog signaling. *Developmental biology*. 194:166-171.
- Iwashita, M., M. Watanabe, M. Ishii, T. Chen, S.L. Johnson, Y. Kurachi, N. Okada, and S. Kondo. 2006. Pigment pattern in jaguar/obelix zebrafish is caused by a Kir7.1 mutation: implications for the regulation of melanosome movement. *PLoS genetics*. 2:e197.
- Kim, B.B., H. Wu, Y.A. Hao, M. Pan, M. Chavarha, Y. Zhao, M. Westberg, F. St-Pierre, J.C. Wu, and M.Z. Lin. 2022. A red fluorescent protein with improved monomericity enables ratiometric voltage imaging with ASAP3. *Scientific reports*. 12:3678.
- Kirschstein, T., M. Rehberg, R. Bajorat, T. Tokay, K. Porath, and R. Kohling. 2009. High K⁺-induced contraction requires depolarization-induced Ca²⁺ release from internal stores in rat gut smooth muscle. *Acta pharmacologica Sinica*. 30:1123-1131.
- Kozak, J.A., and J.W. Putney. 2017. Calcium entry channels in non-excitable cells. CRC Press, Taylor & Francis Group, Boca Raton. xiv, 327 pages pp.
- Matsuoka, S., K. Iwamoto, D.Y. Shin, and M. Ueda. 2024. Spontaneous signal generation by an excitable system for cell migration. *Front Cell Dev Biol*. 12:1373609.
- Nakajima, K.I., K. Zhu, Y.H. Sun, B. Hegyi, Q. Zeng, C.J. Murphy, J.V. Small, Y. Chen-Izu, Y. Izumiya, J.M. Penninger, and M. Zhao. 2015. KCNJ15/Kir4.2 couples with polyamines to sense weak extracellular electric fields in galvanotaxis. *Nature communications*. 6:8532.
- Rienecker, K.D.A., R.G. Poston, and R.N. Saha. 2020. Merits and Limitations of Studying Neuronal Depolarization-Dependent Processes Using Elevated External Potassium. *ASN Neuro*. 12:1759091420974807.
- Silic, M.R., Q. Wu, B.H. Kim, G. Golling, K.H. Chen, R. Freitas, A.A. Chubykin, S.K. Mittal, and G. Zhang. 2020. Potassium Channel-Associated Bioelectricity of the Dermomyotome Determines Fin Patterning in Zebrafish. *Genetics*. 215:1067-1084.
- Ueda, T., K. Terayama, K. Kurihara, and Y. Kobatake. 1975. Threshold phenomena in chemoreception and taxis in slime mold *Physarum polycephalum*. *J Gen Physiol*. 65:223-234.
- Van Duijn, B., S.A. Vogelzang, D.L. Ypey, L.G. Van der Molen, and P.J. Van Haastert. 1990. Normal chemotaxis in *Dictyostelium discoideum* cells with a depolarized plasma membrane potential. *Journal of cell science*. 95 (Pt 1):177-183.
- Villette, V., M. Chavarha, I.K. Dimov, J. Bradley, L. Pradhan, B. Mathieu, S.W. Evans, S. Chamberland, D. Shi, R. Yang, B.B. Kim, A. Ayon, A. Jalil, F. St-Pierre, M.J. Schnitzer, G. Bi, K. Toth, J. Ding, S. Dieudonne, and M.Z. Lin. 2019. Ultrafast Two-Photon Imaging of a High-Gain Voltage Indicator in Awake Behaving Mice. *Cell*. 179:1590-1608 e1523.
- Xiong, Y., C.H. Huang, P.A. Iglesias, and P.N. Devreotes. 2010. Cells navigate with a local-excitation, global-inhibition-biased excitable network. *Proceedings of the National Academy of Sciences of the United States of America*. 107:17079-17086.

September 23, 2025

RE: JCB Manuscript #202503037R

Qing Deng
Purdue University West Lafayette

Dear Dr. Deng:

Thank you for submitting your revised manuscript entitled "Inwardly rectifying potassium channels promote directional sensing during neutrophil chemotaxis". We would be happy to publish your paper in JCB pending final revisions necessary to meet our formatting guidelines (see details below).

In your final revision, please be sure to address reviewer #1's and #3's final minor concerns as well as the caveats about the ASAP3 probe.

A. MANUSCRIPT ORGANIZATION AND FORMATTING:

Full guidelines are available on our Instructions for Authors page, <http://jcb.rupress.org/submission-guidelines#revised>.

1) Text limits: Character count for Reports is < 20,000, not including spaces. Count includes abstract, introduction, results, discussion, and acknowledgments. Count does not include title page, figure legends, materials and methods, references, tables, or supplemental legends.

2) Figures limits: Reports may have up to 5 main text figures.

****3) Figure formatting:** Scale bars must be present on all microscopy images, including inset magnifications. Molecular weight or nucleic acid size markers must be included on all gel electrophoresis. Please include molecular weight markers in Figure 5H. Aspect ratios of images may not be altered.

4) Statistical analysis: Error bars on graphic representations of numerical data must be clearly described in the figure legend. The number of independent data points (n) represented in a graph must be indicated in the legend. Statistical methods should be explained in full in the materials and methods. For figures presenting pooled data the statistical measure should be defined in the figure legends. Please also be sure to indicate the statistical tests used in each of your experiments (either in the figure legend itself or in a separate methods section) as well as the parameters of the test (for example, if you ran a t-test, please indicate if it was one- or two-sided, etc.). Also, if you used parametric tests, please indicate if the data distribution was tested for normality (and if so, how). If not, you must state something to the effect that "Data distribution was assumed to be normal but this was not formally tested."

5) Abstract and title: The abstract should be no longer than 160 words and should communicate the significance of the paper for a general audience. The title should be less than 100 characters including spaces. Make the title concise but accessible to a general readership.

****6) Materials and methods:** Should be comprehensive and not simply reference a previous publication for details on how an experiment was performed. Please provide full descriptions in the text for readers who may not have access to referenced manuscripts. For example, in the "Generation of Transgenic and Mutant Zebrafish Lines" paragraph of your methods section, you state: "The stable lines were generated as previously described (Deng et al., 2011)". You must remove this and write out the full methodology. The same goes for all other places in the methods where you refer to methods "...as previously described."

7) All antibodies, cell lines, animals, and tools used in the manuscript should be described in full, including accession numbers for materials available in a public repository such as the Resource Identification Portal. Please be sure to provide the sequences for all of your primers/oligos and RNAi constructs in the materials and methods. You must also indicate in the methods the source, species, and catalog numbers (where appropriate) for all of your antibodies. Please also indicate the acquisition and quantification methods for immunoblotting/western blots.

8) Microscope image acquisition: The following information must be provided about the acquisition and processing of images:

- Make and model of microscope
- Type, magnification, and numerical aperture of the objective lenses
- Temperature

- d. Imaging medium
- e. Fluorochromes
- f. Camera make and model
- g. Acquisition software
- h. Any software used for image processing subsequent to data acquisition. Please include details and types of operations involved (e.g., type of deconvolution, 3D reconstitutions, surface or volume rendering, gamma adjustments, etc.).

10) Supplemental materials: There are strict limits on the allowable amount of supplemental data. Reports may have up to 3 supplemental figures. Please also note that tables, like figures, should be provided as individual, editable files. A summary of all supplemental material should appear at the end of the Materials and methods section.

**13) ORCID IDs: ORCID IDs are unique identifiers allowing researchers to create a record of their various scholarly contributions in a single place. Please note that ORCID IDs are now *required* for all authors. At resubmission of your final files, please be sure to provide your ORCID ID and those of all co-authors.

Please note that JCB now requires authors to submit Source Data used to generate figures containing gels and Western blots with all revised manuscripts. This Source Data consists of fully uncropped and unprocessed images for each gel/blot displayed in the main and supplemental figures. For assays performed using capillary electrophoresis and/or immunoassay-based detection, authors should instead provide the electropherogram graph(s) for each experiment, plotting fluorescence/chemiluminescence intensity vs. molecular weight/size. Please be sure to provide one Source Data file for each figure gels, blots, and/or capillary electrophoresis assays along with your revised manuscript files. File names for Source Data figures should be alphanumeric without any spaces or special characters (i.e., SourceDataF#, where F# refers to the associated main figure number or SourceDataFS# for those associated with Supplementary figures). For traditional gels and blots, the lanes of the gels/blots should be labeled as they are in the associated figure, the place where cropping was applied should be marked (with a box), and molecular weight/size standards should be labeled wherever possible. For capillary electrophoresis assays, each trace in the graph should be color-coded and labeled to indicate which protein, gene, or sample is being measured (please try to avoid red/green combinations to accommodate our color-blind readers).

Journal of Cell Biology now requires a data availability statement for all research article submissions. These statements will be published in the article directly above the Acknowledgments. The statement should address all data underlying the research presented in the manuscript. Please visit the JCB instructions for authors for guidelines and examples of statements at (<https://rupress.org/jcb/pages/editorial-policies#data-availability-statement>).

B. FINAL FILES:

-- High-resolution figure and MP4 video files: See our detailed guidelines for preparing your production-ready images,

<https://jcb.rupress.org/fig-vid-guidelines>.

Thank you for your attention to these final processing requirements. Please revise and format the manuscript and upload materials within 7 days. If you need an extension for whatever reason, please let us know and we can work with you to determine a suitable revision period.

Thank you for this interesting contribution, we look forward to publishing your paper in Journal of Cell Biology.

Sincerely,

Anna Huttenlocher
Monitoring Editor
Journal of Cell Biology

Gabriele Stephan
Scientific Editor
Journal of Cell Biology

Reviewer #1:

Major remaining comment:

I am still skeptical about the ratiometric quantification of the ASAP3 probe with CAAX, which still does not convince me. Although I am not an absolute expert in the detailed methodology of this probe, I started to inform myself in the literature a bit more. Unfortunately, the reviewer's answers still do not provide a fully conclusive proof that their probe is calibrated to the membrane potential. Since the authors make such a strong statement about the maintenance of a membrane potential gradient in the cell over time, the authors - in their own interest - have to make sure that they are later not proven wrong by other researchers. How would such a gradient be maintained over time? Electrophysiologists would argue that any potential difference would be balanced within milliseconds.

The reviewers give many reasons why calibration experiments are difficult (which I understand). They mention that calibration experiments in zebrafish are technically difficult (which I can see), and their cellular system (dHL60 cells) expresses the ASAP3 probe in a weird fashion (which is unfortunate, given that here calibration experiments would be possible). They refer for calibration measurements to a paper from Peter Devreotes (Banerjee et al., 2022) in another cell type, where measurements of a flat distribution of fixed charges on the inside of a membrane ("horizontal") have been performed. However, the membrane potential is an electrical gradient over the membrane ("vertical").

Currently, it would still be possible that the authors detect a pH gradient within the cell. GFP is very sensitive to pH, which is not the case for mCherry. There are no controls to rule that out, i.e. pH sensitivity of the fluorescent probes.

The best probe and methodology for addressing the issue here appears to be "rEstus", mentioned in my previous review and

then also referred to by the authors (doi: 10.1002/advs.202307938). Here there is also a methodology which does not use the line scan across the cell for quantifying the membrane potential. However, the authors say that neutrophils would be too fast for using this probe and methodology - which I am actually doubting.

I really do not want to block the publication of this manuscript, but - in their own interest - the authors should make every effort to not be proven wrong later due to technical issues, which would hurt them scientifically in the long run.

Other remaining comments:

1) Many authors might wonder why the ASAP3 probe has not been used with the dHL60 cells. The authors should at least mention in the Material and Methods section that ASAP3 expression in dHL60 cells gives endosomal localization (as explained to the reviewer) that others understand why this experiment has not been performed and presented.

2) It is very unclear to me why the under-agarose chemotaxis assay of primary human neutrophils and dHL60 cells has been performed so differently. The experimental design of the under-agarose assay is pretty uncommon (agarose is overlaid on top of the seeded cells), including the huge number of cells that "stick" to the agarose. It remains unclear why the authors used this modified technique, which also only results in the chemotactic movement of a small fraction of cells. Not entirely convincing.

Reviewer #2:

My comments have been adequately addressed.

Reviewer #3:

The authors have adequately responded to my previous criticisms and questions and appropriately revised the manuscript. In particular, the clarified quantification of ASAP3 data, the added pharmacology and genetic controls strengthening VU590 specificity, and the inclusion of primary mouse neutrophil data all substantially improve clarity and confidence in the conclusions.

Furthermore, I appreciate the authors' attempts at in vivo patch-clamping, even if these were not ultimately successful. A few minor edits and corrections remain in the manuscript:

1, ensure consistent wording that ASAP3 brightness decreases (and not increases) upon depolarization across the text and figure captions (this was also inconsistently described in the rebuttal text)

2, check for typos e.g. "10 M KCl" (likely 10 mM - line 441), and also the one I already pointed out in the previous review: "to investigate where Kir regulates neutrophil migration" (line121)

Aside from these minor points, I recommend the manuscript for publication.

We thank the reviewers for endorsing our manuscript for publication. We have addressed the remaining concerns and updated the text accordingly. Please see our point-by-point response below with the comments in bold.

Reviewer 1:

Major remaining comment:

I am still skeptical about the ratiometric quantification of the ASAP3 probe with CAAX, which still does not convince me. Although I am not an absolute expert in the detailed methodology of this probe, I started to inform myself in the literature a bit more. Unfortunately, the reviewer's answers still do not provide a fully conclusive proof that their probe is calibrated to the membrane potential. Since the authors make such a strong statement about the maintenance of a membrane potential gradient in the cell over time, the authors - in their own interest - have to make sure that they are later not proven wrong by other researchers. How would such a gradient be maintained over time? Electrophysiologists would argue that any potential difference would be balanced within milliseconds.

We appreciate the reviewer's concern about the strong statement regarding maintaining a membrane potential gradient in the cell over time. We have revised the statement to be less absolute, as we cannot exclude all possible artifacts. We have added a possible explanation in the discussion for how the potential difference might be maintained during active protrusion. We speculate that positive feedback at the cell front is necessary for the gradient to form.

The reviewers give many reasons why calibration experiments are difficult (which I understand). They mention that calibration experiments in zebrafish are technically difficult (which I can see), and their cellular system (dHL60 cells) expresses the ASAP3 probe in a weird fashion (which is unfortunate, given that here calibration experiments would be possible). They refer for calibration measurements to a paper from Peter Devreotes (Banerjee et al., 2022) in another cell type, where measurements of a flat distribution of fixed charges on the inside of a membrane ("horizontal") have been performed. However, the membrane potential is an electrical gradient over the membrane ("vertical").

We have data showing that VU590 and CoCHR activation cause a change in the ASAP3/mCherryCAAX ratio, which confirms that ASAP3 can measure relative membrane potential (V_m) in zebrafish neutrophils. We tried a calibration experiment to determine the exact V_m at both the front and back of the cell, but it was not successful. In the future, we may incorporate FLIM measurement, which could provide an absolute V_m reading (doi: <https://doi.org/10.1101/2025.08.08.669310>). This technique will likely need significant optimization to work with fast-moving cells in tissue. The localization of ASAP3 and mCherry CAAX overlaps perfectly, indicating they are both on the plasma membrane. 3D migration is more complex than 2D—the cell front is not always on the x-y plane. Therefore, analyzing the cell outline on the z-projection has limitations, as it does not precisely show the cell front.

Analyzing cell polarity using a line scan across the cell body helps identify a broader region as the cell front, improving accuracy. Ideally, cell shape changes should be analyzed in 3D, but instrument limitations in z-resolution and the computational pipeline make this difficult. Nonetheless, the line scan method is commonly used in cell migration studies and should not introduce additional artifacts.

Currently, it is still possible that the authors detect a pH gradient inside the cell. GFP is very sensitive to pH, unlike mCherry. There are no controls to rule this out, such as testing the pH sensitivity of the fluorescent probes.

We respectfully disagree with the reviewer that the gradient is unlikely due to a pH gradient. The activity of NHE1 at the cell front (H⁺ exit and Na⁺ entry) will make the cell front more alkaline, which promotes actin polymerization. GFP family sensors decrease intensity in more acidic environments. Therefore, reduced ASAP3 intensity at the cell front is unlikely to be an artifact of a pH gradient.

The best probe and methodology for addressing the issue here appears to be "rEstus", mentioned in my previous review and then also referred to by the authors (doi: 10.1002/adv.202307938). Here there is also a methodology which does not use the line scan across the cell for quantifying the membrane potential. However, the authors say that neutrophils would be too fast for using this probe and methodology - which I am actually doubting.

We use confocal microscopy to capture 3D migration. We turn on both lasers simultaneously to acquire data from both channels. We balanced the speed needed for resolution and cell movements. It took 15 seconds to complete one z-stack scan of neutrophils. We observe significant cell movement after each scan. Therefore, we did not attempt rEstus, which requires two different excitation wavelengths and sequential acquisition.

I really do not want to block the publication of this manuscript, but - in their own interest - the authors should make every effort to not be proven wrong later due to technical issues, which would hurt them scientifically in the long run.

We thank the reviewer for the suggestion. We are carefully drawing conclusions based on our current data. We have rephrased our conclusion to lessen its absoluteness and acknowledged the potential artifacts we cannot fully eliminate.

Other remaining comments:

1) Many authors might wonder why the ASAP3 probe has not been used with the dHL60 cells. The authors should at least mention in the Material and Methods section that ASAP3 expression in dHL60 cells gives endosomal localization (as explained to the reviewer) that others understand why this experiment has not been performed and presented.

We have indicated in the main text that ASAP3 expression in dHL-60 cells is trapped in intracellular compartments.

2) It is very unclear to me why the under-agarose chemotaxis assay of primary human neutrophils and dHL60 cells has been performed so differently. The experimental design of the under-agarose assay is pretty uncommon (agarose is overlaid on top of the seeded cells), including the huge number of cells that "stick" to the agarose. It remains unclear why the authors used this modified technique, which also only results in the chemotactic movement of a small fraction of cells. Not entirely convincing.

We updated our *under-agarose migration assay* because the new approach offers higher reproducibility and reduces downstream analytical complexity.

Initially, we performed the assay by casting an agarose gel onto collagen-coated 35 mm Petri dishes, punching two 2 mm holes spaced 2.5 mm apart with a metal tube, loading cells and chemoattractant into the respective wells, and imaging with a BioTek Lionheart Benchtop Automated Microscope. However, the original method had several technical limitations:

1. Sample throughput — due to imaging speed constraints, the Lionheart microscope could handle only 2–4 samples at a time, limiting efficiency and consistency during revision work.
2. Hole preparation — punching holes often damaged the agarose or left gel residue that blocked cell migration, occasionally resulting in no cells migrating out of the well.
3. Evaporation and gradient instability — the small wells lost fluid during long-term imaging, altering chemoattractant concentration and reducing image quality (lower liquid level caused poor focus and cell contour visibility).
4. Cell crowding — cells loaded into tiny wells tended to cluster and cross migration paths, making it harder to track individual migration trajectories reliably.

To overcome these issues, we adopted an improved design using ibidi 8-well chamber slides instead of 35 mm dishes. This allowed simultaneous imaging of up to eight samples and eliminated the need for hole punching. In each well, we created a chemoattractant reservoir 2.5 mm wide along the right wall. Cells remained under the gel throughout migration, preventing agarose debris from obstructing movement and enabling a more controlled chemoattractant gradient. Larger chemoattractant volumes reduced evaporation-induced concentration changes. The chamber slides also enabled us to use higher-resolution spinning-disk confocal microscopy for live imaging. Additionally, by seeding cells before adding agarose, we improved cell dispersion, reduced path crossing, and enhanced tracking accuracy.

We appreciate the reviewer's comment regarding "only a small fraction of cells migrating." In early optimization, we observed most cells were highly motile under the gel, and this setup can also reveal chemokinesis or random migration. Unfortunately, during revision, our confocal microscope's humidity control system failed and could not be repaired in time, so we had to use

moist Kimwipe tissue for humidity. This imperfect control caused our 1.5 % agarose gels to dry faster than usual, altering gel stiffness and increasing mechanical resistance, resulting in fewer migrating cells. Nevertheless, although the overall migration ratio of input neutrophils was reduced, the treatment-dependent differences remained robust and reproducible. We therefore believe the improved *under-agarose migration assay* remains valid and reliable for our study.

Reviewer #3:

The authors have adequately responded to my previous criticisms and questions and **1, ensure consistent wording that ASAP3 brightness decreases (and not increases) upon depolarization across the text and figure captions (this was also inconsistently described in the rebuttal text)**

We thank the reviewer for pointing it out. We have updated the text for accuracy.

2, check for typos e.g. "10 M KCl" (likely 10 mM - line 441), and also the one I already pointed out in the previous review: "to investigate where Kir regulates neutrophil migration" (line121)

We thank the reviewer for pointing it out. We have removed the concentrations here, as they are specified in the other sections of the methods.